# Mutual information and task-relevant latent dimensionality

Paarth Gulati[1, *], Eslam Abdelaleem[2, *], Audrey Sederberg[2] and Ilya Nemenman[1,3]

[1]Department of Physics, Initiative for Theory and Modeling of Living Systems, Emory University
[2]School of Physics, School of Psychology, Georgia Institute of Technology
[3]Department of Biology, Emory University

## Abstract

Estimating the dimensionality of the latent representation needed for prediction—the task-relevant dimension—is a difficult, largely unsolved problem with broad scientific applications. We cast it as an Information Bottleneck question: what embedding bottleneck dimension is sufficient to compress predictor and predicted views while preserving their mutual information (MI). This repurposes neural MI estimators for dimensionality estimation. We show that standard neural estimators with separable/bilinear critics systematically inflate the inferred dimension, and we address this by introducing a hybrid critic that retains an explicit dimensional bottleneck while allowing flexible nonlinear cross-view interactions, thereby preserving the latent geometry. We further propose a one-shot protocol that reads off the effective dimension from a single over-parameterized hybrid model, without sweeping over bottleneck sizes. We validate the approach on synthetic problems with known task-relevant dimension. We extend the approach to intrinsic dimensionality by constructing paired views of a single dataset, enabling comparison with classical geometric dimension estimators. In noisy regimes where those estimators degrade, our approach remains reliable. Finally, we demonstrate the utility of the method on multiple physics datasets.

## 1 Introduction

Before "low-dimensional latent embeddings" became a rallying cry of AI, they were already a basic aim of science: identify a low-dimensional *state*—a small set of degrees of freedom constructed from observations—that suffices to predict the quantities of interest. The long road from Aristotelian to Newtonian mechanics illustrates that determining the *number* of such state variables—the relevant *latent dimensionality*—can be hard, even before one argues about the right variables or the laws that relate them. In today's high-throughput, AI-enabled scientific world, the (somewhat Wignerian) "unreasonable effectiveness" (Wigner, 1960) of low-dimensional descriptions in modeling complex physical systems is reinforcing the view that such latent structure is often a property of the underlying systems rather than a modeling choice (Huh et al., 2024; Edamadaka et al., 2025). The evidence for this spans fluid dynamics (Chen et al., 2022), molecular structure and dynamics (Das et al., 2006; Tamura et al., 2022), jammed materials (Cubuk et al., 2017; Jin & Yoshino, 2021), and neural population activity (Gallego et al., 2017; Semedo et al., 2019; Schneider et al., 2023). Yet estimates of the *relevant* latent dimensionality often vary widely across methods, even within the same study.

This problem has two roots. First, for noisy scientific data an "intrinsic" dimensionality of the raw observations is neither unique nor especially useful: for example, the variables

---

[*]P.G. and E.A. contributed equally to this work.

needed to predict a body's future position are not the same as those needed to record its shape. For scientific applications, one needs instead a *task-relevant* dimension—the size of the minimal state required to preserve the information relevant for a specified prediction problem. While there are niche attempts to adapt intrinsic dimension notions to such task-relevant settings, there is no accepted general solution. Second, even if one asks only for an intrinsic dimension of the data distribution, estimation is notoriously fragile in the high-dimensional, undersampled regime common in science. Both classical nonlinear-dynamics estimators (Grassberger & Procaccia, 1983) and modern neighbor-statistics methods such as Levina–Bickel (Levina & Bickel, 2004) and Two-NN (Facco et al., 2017) (see also (Camastra & Staiano, 2016)) can return convincing but meaningless estimates on limited, noisy data (Theiler, 1986). More fundamentally, for realistic dataset sizes these approaches often saturate at dimensions "like 6 or 7" (Eckmann & Ruelle, 1992)—a vintage ancestor of a modern meme.

In this paper, we use mutual information (MI) to formalize task relevance. We leverage recent advances in neural MI estimation (Belghazi et al., 2018; van den Oord et al., 2018; Poole et al., 2019; Song & Ermon, 2019) to estimate the dimensionality. Given paired views $(X, Y)$—a predictor and the quantity to be predicted—we seek the smallest bottleneck dimension $k_z$ for which compressed representations $Z_X = f(X) \in \mathbb{R}^{k_z}$ and $Z_Y = g(Y) \in \mathbb{R}^{k_z}$ preserve the shared information, i.e., $I(Z_X; Z_Y) \approx I(X; Y)$. This symmetric information bottleneck (SIB) viewpoint (Friedman et al., 2013; Abdelaleem et al., 2025b) retains only cross-view dependencies (no data reconstruction) and can, therefore, be markedly more data-efficient (Martini & Nemenman, 2024; Van Assel et al., 2025), reducing the sample demands that limit classical estimators.

We show, analytically and experimentally, that popular separable/bilinear critics (van den Oord et al., 2018) can systematically *overestimate* task-relevant dimensionality in this pipeline, even for simple latent distributions. We mitigate this by introducing a *hybrid critic* that retains an explicit $k_z$ bottleneck while allowing flexible cross-view mixing, which we find is essential for capturing nonlinear dependence geometry without inflating the embedding dimension. We also give a protocol that reads off the effective dimension from a single over-parameterized hybrid model, avoiding a sweep over $k_z$. For finite datasets we use the max-test early-stopping rule (Abdelaleem et al., 2025a) and validate the method on teacher problems with known task-relevant dimension. To enable comparison with intrinsic-dimension estimators, we extend our approach to an intrinsic setting by splitting a single dataset into two equivalent views. In noisy regimes where classical geometric estimators degrade, the hybrid MI/SIB approach remains reliable. Finally, we demonstrate utility on realistic physical data by applying the estimator to 2D Ising spins simulations (recovering critical scaling) and to single/double pendulum videos (recovering phase-space dimensionality).

**Contributions.** (i) Task-relevant dimensionality from paired views as a symmetric MI preservation/SIB problem; (ii) an analytic and empirical demonstration of dimensionality inflation induced by separable/bilinear critics; (iii) a hybrid critic and one-shot effective-dimension estimator with a finite-data training protocol; (iv) validation on teacher benchmarks, an intrinsic-dimension extension via view splitting, and robustness to observation noise; (v) applications to physics datasets.

## 2  SETUP AND METHODOLOGY

We will use mutual information (MI) between two variables $X$ and $Y$ as a measure of task relevance:

$$I(X; Y) = \mathbb{E}_{p(x)}[D_{\mathrm{KL}}(p(y|x) \,\|\, p(y))] = I(Y; X). \tag{1}$$

Using the Donsker–Varadhan (DV) representation of the KL divergence (Donsker & Varadhan, 1983), $D_{\mathrm{KL}}(P\|Q) = \sup_{T:\Omega\to\mathbb{R}} \mathbb{E}_P[T] - \log\big(\mathbb{E}_Q[e^T]\big)$, where the supremum is taken over measurable functions sharing the common support $\Omega$ of $P$ and $Q$, one obtains a variational lower bound on MI: with expectations approximated by finite-sample averages, one trades estimation of $I$ for optimization over a critic function $T(x, y)$, implemented with sufficiently expressive neural networks (Belghazi et al., 2018). In this work, we largely focus on the

*symmetrized* InfoNCE estimator (van den Oord et al., 2018; Radford et al., 2021), which is derived from DV-style bounds by *contrastive* averaging. It approaches the true MI when $T$ equals the optimal critic $T^*(x, y) = \log\big(p(x, y)/p(x)p(y)\big)$, provided the number of negatives and the batch size are large. See App. A.1 and Abdelaleem et al. (2025a) for a comparison of neural MI objectives, critics, and architectures. While InfoNCE is a strong default, the exact choice of the estimator is not crucial to the rest of the story.

MI estimators also differ in how the critic is parameterized. The simplest choice is a *concatenated (joint)* critic, which trains a single network $T_{\text{concat}} : \mathcal{X} \times \mathcal{Y} \to \mathbb{R}$ (Belghazi et al., 2018). Another common choice is a *separable* critic (van den Oord et al., 2018), which trains two encoders $g^X : \mathcal{X} \to \mathbb{R}^{k_z}$ and $g^Y : \mathcal{Y} \to \mathbb{R}^{k_z}$ and represents the critic as the bilinear form (dot product) $T_{\text{sep}}(x, y) = g^X(x) \cdot g^Y(y)$. Unlike the concatenated critic, the separable form introduces an explicit $k_z$-dimensional bottleneck, making it natural to study how the estimate depends on representation size. With sufficiently expressive networks (and appropriate training protocols), either family can yield accurate MI estimates in high dimensions (Abdelaleem et al., 2025a).

We note that symmetrized InfoNCE is widely used and performs well in modern representation learning (Chen et al., 2020; Radford et al., 2021). We therefore adopt this objective throughout. Our **proposal** for estimating task-relevant dimensionality is then straightforward: use symmetrized InfoNCE with an embedding bottleneck and finite-data improvements that yield nearly unbiased estimates (Abdelaleem et al., 2025a)[1], sweep over the bottleneck dimension $k_z$, and identify the smallest $k_z$ beyond which the MI estimate no longer increases within error bars as the task-relevant dimension. Crucially, we will show that existing critic architectures fail in this pipeline: a concatenated critic has no explicit bottleneck to interpret as task-relevant dimension, while a separable critic can require *inflated* $k_z$ to represent nonlinear dependencies, leading to systematic overestimation. We therefore introduce a *hybrid* critic,

$$T_{\text{hybrid}}(x, y) = T_\theta\big([g^X(x), g^Y(y)]\big), \qquad (2)$$

Figure 1: **Hybrid Critic Architecture.** Retains the bottleneck for dimensionality analysis, but allows flexible mixing via a concatenated head $T_\theta$ (e.g. a small MLP).

which retains the $k_z$ bottleneck but uses a lightweight network $T_\theta : \mathbb{R}^{k_z} \times \mathbb{R}^{k_z} \to \mathbb{R}$ to capture nonlinear cross-view interactions without inflating $k_z$ (Figure. 1). This decouples *representation size* from *critic expressivity*, enabling the learned embeddings to reflect the shared task-relevant latent dimensionality (and, as we show later, intrinsic dimensionality) without increasing $k_z$. Parenthetically, the hybrid architecture also performs well as an MI estimator in its own right.

In principle, $T_\theta$ can be a small Multilayer Perceptron (MLP) head operating in the latent space, while the encoders can be tailored to the modality (e.g., incorporating invariances or convolutional structure) to map observations into a shared latent space. Here we instead fix both the encoders and the hybrid head to the same small MLP architecture across all experiments, and show that this minimal design still recovers shared latent and intrinsic dimensionality across diverse datasets where existing estimators fail.

## 3   RESULTS

We first present estimation of task-relevant dimensionality on synthetically generated datasets constructed from low-dimensional ($\mathcal{O}(1)$) latent variables mapped into a high-dimensional observation space by fixed nonlinear functions $F$ (see App. E.1). We use the following notation: $Z_X$ and $Z_Y$ are latent variables with joint distribution $p_Z(z_x, z_y)$ and dimensions $K_{Z_X} = \dim(Z_X)$ and $K_{Z_Y} = \dim(Z_Y)$. Observations are $(X, Y) =$

---

[1] See also Assran et al. (2023); Monemi et al. (2025) for connections to joint-embedding architectures.

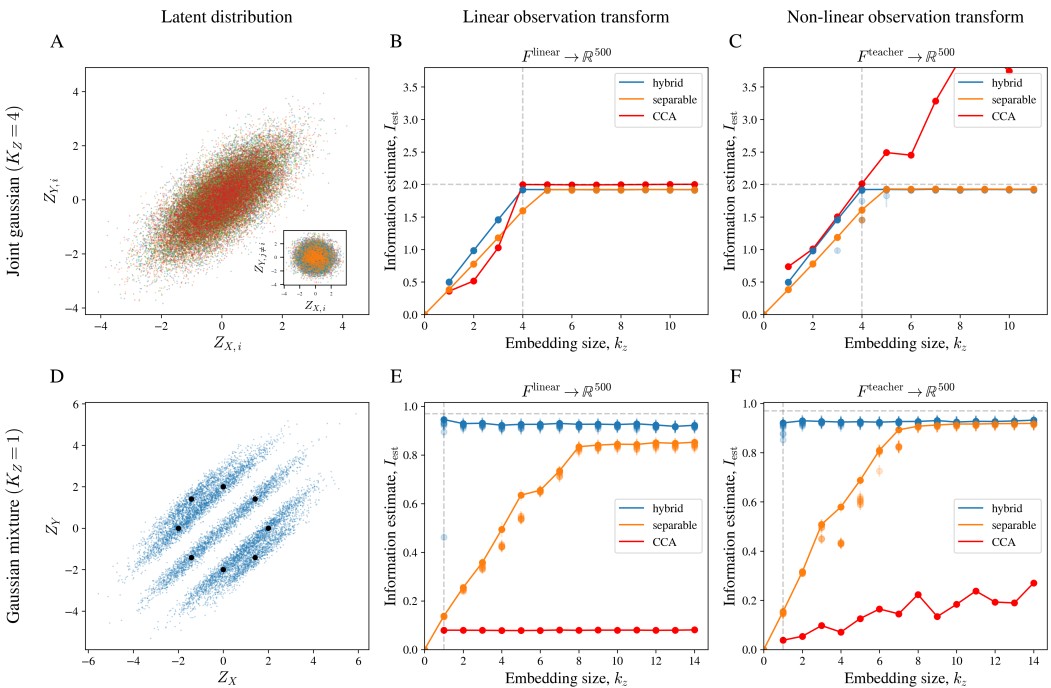

Figure 2: **Role of embedding size in the infinite-data (resampling) regime.** Estimated MI versus encoder embedding size $k_z$ for two latent distributions: **(A–C)** jointly Gaussian latent ($K_Z = 4$) with total MI $I = 2.0$ bits (equal per latent dimension); **(D–F)** Gaussian mixture with $N_p = 8$ equally likely joint-Gaussian clusters with $k_Z = 1$ (each with $\rho \approx 0.97$), with cluster means on a circle of radius $\mu = 2.0$ (see App. E.1). **(A,D)** Latent distributions. **(B,E)** MI estimates (maximum over 10 trials, individual trials shown with semi-transparent markers) for frozen linear observation maps. **(C,F)** Same, but with frozen nonlinear teacher maps (see App. E.1). Vertical dotted lines mark true task-relevant dimension, which is always matched by $k_z^*$ chosen by the hybrid critic.

$(F_X(Z_X), F_Y(Z_Y))$, with $F_X : \mathbb{R}^{K_{Z_X}} \to \mathbb{R}^{K_X}$ and $F_Y : \mathbb{R}^{K_{Z_Y}} \to \mathbb{R}^{K_Y}$. Unless otherwise stated, we take $K_{Z_X} = K_{Z_Y} \equiv K_Z$ and $K_X = K_Y \equiv K$.

We focus on the physically relevant regime $K \gg K_Z$ and aim to infer $K_Z$ from samples of the $K$-dimensional observations $(X, Y)$. Unless otherwise stated, we fix $K = 500$ and vary $K_Z$, the latent distribution $p_Z$, and the observation maps $F$, while comparing critic sizes and architectures.

Across all considered benchmarks, the *hybrid* architecture reliable infers the latent dimensionality. We therefore apply the same pipeline to more realistic data, with similarly good results.

### 3.1 Infinite Data Regime: high dimensional input, low dimensional latents

We begin in the *infinite-data* regime, where each optimization step uses an independently sampled batch from the data-generating distribution. This eliminates overfitting and isolates the estimator's intrinsic behavior, rather than finite-sample effects. To use MI as a dimensionality estimator, we study the optimized MI estimate as a function of an effective representation size. For neural estimators this size is the embedding dimension $k_z$ of the encoders $g_X, g_Y : \mathbb{R}^K \to \mathbb{R}^{k_z}$ (we use same dimensionality encoders throughout). We also investigate MI estimation via CCA (Gelfand, 1959) (also see App. A.2.1), where the analogous knob is the number of retained canonical pairs.

For sufficiently large $k_z$, the MI estimate generically saturates to the true MI of the latent distribution. We denote the saturation location by $k_z^*$. In general, $k_z^*$ depends on the latent distribution $p_Z$, the true latent dimensionality $K_Z$, and the estimator/critic architecture.

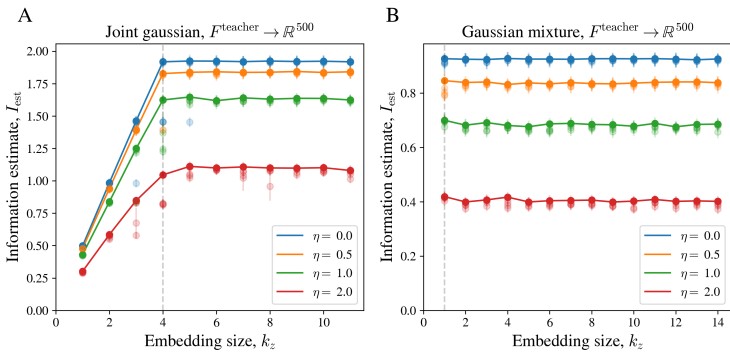

Figure 3: **Effect of additive observation noise (hybrid critic).** Independent white noise is added after the frozen nonlinear observation map, with $\langle \eta_{\alpha,i}\eta_{\beta,j} \rangle = \sigma_\alpha^2 \delta_{ij}\delta_{\alpha\beta}$ and strength set by the noise-to-signal ratio $\eta$. **(A)** Joint Gaussian latent ($K_Z = 4$). **(B)** Gaussian mixture ($N_p = 8$ components). Noise reduces the estimated MI, while the saturation point used to infer dimensionality is preserved. MI is the maximum over 10 trials; individual trials shown as semi-transparent markers.

Figure 2 illustrates this behavior for two representative latent distributions (many more were tested) used throughout this section: (i) a jointly Gaussian latent with total MI $I = 2.0$ bits and true shared latent dimensionality $K_Z = 4$, Fig. 2A, and (ii) a deliberately challenging, multimodal Gaussian mixture with $K_Z = 1$ and multiple equally likely clusters, Fig. 2D. In each case we generate observations using either frozen linear maps or frozen nonlinear teacher networks.

In the infinite-data regime, CCA succeeds only for the jointly Gaussian latent with linear observations (Fig. 2B). With nonlinear observation maps, CCA picks up spurious correlations and fails to recover both MI and latent dimensionality in both benchmarks (Fig. 2C,E,F). The *separable* critic can estimate MI but does not saturate at the correct dimensionality: for the joint Gaussian, saturation occurs at $k_z^* = K_Z + 1$ (Fig. 2B,C; App. A.2), while the mixture yields substantial overestimation (Fig. 2E,F; App. D.1). In contrast, for the *hybrid* critic we find $k_z^* = K_Z$ across both latent distributions and both observation transforms, making $k_z^*$ a reliable estimate of the shared task-relevant latent dimensionality. We therefore focus on the hybrid critic in the remainder of this work.

## 3.2 Noise in the observation space and intrinsic dimensionality

We target our estimator to scientific applications, where data are always noisy. We thus test robustness to measurement noise by adding observation noise after the (frozen) observation maps:

$$X = F_X(Z_X) + \eta_X, \qquad Y = F_Y(Z_Y) + \eta_Y, \tag{3}$$

with $\eta_X, \eta_Y$ uncorrelated Gaussian white noise, $\langle \eta_{\alpha,i}\eta_{\beta,j} \rangle = \sigma_\alpha^2 \delta_{ij}\delta_{\alpha\beta}$ for $\alpha, \beta \in \{X, Y\}$ and $i, j \in \{1, \ldots, K\}$. Noise reduces the achievable MI between $X$ and $Y$, and we test whether the saturation point $k_z^*$ remains controlled by the shared latent dimensionality rather than by noise.

Figure 3 shows the optimized MI estimate versus embedding size for the same two representative latent distributions as in Fig. 2. We parameterize the noise level by the noise-to-signal ratio $\eta := \sqrt{\sigma_X^2/\text{var}(F_X(Z_X))} = \sqrt{\sigma_Y^2/\text{var}(F_Y(Z_Y))}$. Increasing noise lowers the estimated MI, but the dimensionality inferred from the saturation point $k_z^*$ is unchanged across the range shown for both latent distributions.

The same behavior holds in a shared-latent setting, where two noisy views are generated from a single latent variable (Appendix C.1, Fig. 10), motivating our later "view-splitting" constructions for intrinsic-dimension benchmarks and real data. For comparison, standard intrinsic-dimension estimators such as Levina–Bickel and Two-NN (Levina & Bickel, 2004; Facco et al., 2017) degrade sharply under observation noise and tend to track the ambient ob-

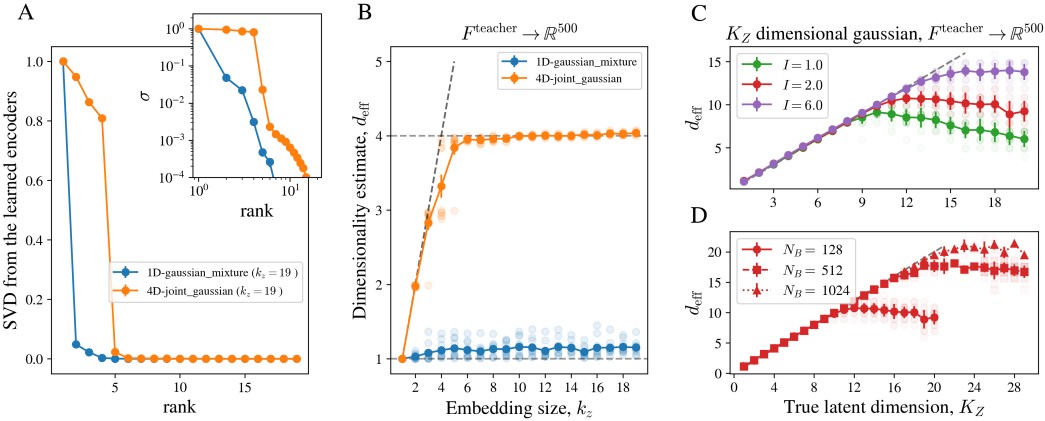

Figure 4: **Single-shot dimensionality from the embedding spectrum (hybrid critic).** The participation ratio of the cross-covariance spectrum of the learned encoder embeddings provides a reliable estimate of latent dimensionality. **(A)** Normalized singular values of the cross-covariance (computed from $10^4$ samples) for a trained model with $k_z = 19$ (inset: log-scale). A clear gap appears after the first $K_Z$ modes, yielding $d_{\mathrm{eff}}$ via Eq. 4. **(B)** $d_{\mathrm{eff}}$ saturates for $k_z \gtrsim K_Z$ for both representative latent distributions, indicating that the learned embeddings concentrate onto an effectively $K_Z$-dimensional subspace. **(C,D)** $d_{\mathrm{eff}}$ from a single over-parameterized model ($k_z = 64$) versus $K_Z$ for jointly Gaussian latents: varying total MI at fixed batch size $N_B = 128$ (C), and varying $N_B$ at fixed total MI $I = 2$ bits (D). As always, semi-transparent markers denote individual trials, and error bars are standard deviations.

servation dimension rather than the latent one (Appendix C.2, Fig. 11); in the same regime, our hybrid critic MI-based estimator continues to recover the shared latent dimensionality.

### 3.3 SINGLE-SHOT DIMENSIONALITY ESTIMATION VIA PARTICIPATION RATIO

So far we inferred latent dimensionality from the saturation of the optimized MI estimate as a function of bottleneck size $k_z$ for the hybrid critic. Sweeping $k_z$ is computationally expensive and often impractical. Empirically, the variability of the learned MI estimates across runs decreases as $k_z$ increases beyond $K_Z$ (cf. Figs. 2E,F and 3A), suggesting more stable optimization under overparameterization. This raises a natural question: when $k_z > K_Z$, do the learned embeddings actually use the additional dimensions, or does the cross-view signal concentrate onto an effectively $K_Z$-dimensional subspace?

We quantify this by the cross-covariance of the encoder outputs,

$$C_{xy} = \left(g_X(X) - \bar{g}_X(X)\right)^T \left(g_Y(Y) - \bar{g}_Y(Y)\right), \qquad \bar{g}_X = \langle g_X \rangle, \;\; \bar{g}_Y = \langle g_Y \rangle,$$

and define an effective dimension from its singular values $\{\sigma_i\}_{i=1}^{k_z}$ via the participation ratio

$$d_{\mathrm{eff}} = \frac{\left(\sum_i \sigma_i\right)^2}{\sum_i \sigma_i^2}. \tag{4}$$

This choice soft-counts dominant modes while suppressing the noise tail (see App. B.2).

Figure 4A shows that for $k_z \gg K_Z$ the embedding spectrum exhibits a clear gap beyond rank $K_Z$ for both representative latent distributions, yielding $d_{\mathrm{eff}}$ from a single trained model. As shown in Fig. 4B, $d_{\mathrm{eff}}$ is stable for $k_z \gtrsim K_Z$, with variance of $d_{\mathrm{eff}}$ across trials decreasing as $k_z$ increases, again reflecting more stable training.

This yields a dimensionality estimator without a $k_z$ sweep: choose $k_z$ sufficiently large, train one hybrid MI estimator, and report $d_{\mathrm{eff}}$ from the embedding spectrum (in practice, take $k_z - d_{\mathrm{eff}} \gg 1$). Figures 4C,D show exact recovery $d_{\mathrm{eff}} = K_Z$ for jointly Gaussian latents with varying $K_Z$ observed through a nonlinear map into $K = 500$ dimensions using a fixed $k_z = 64$. Deviations arise only when finite-batch effects dominate the latent signal, either because the signal per latent dimension is small (Fig. 4C) or because the batch size $N_B$ is insufficient to sample all latent directions (Fig. 4D).

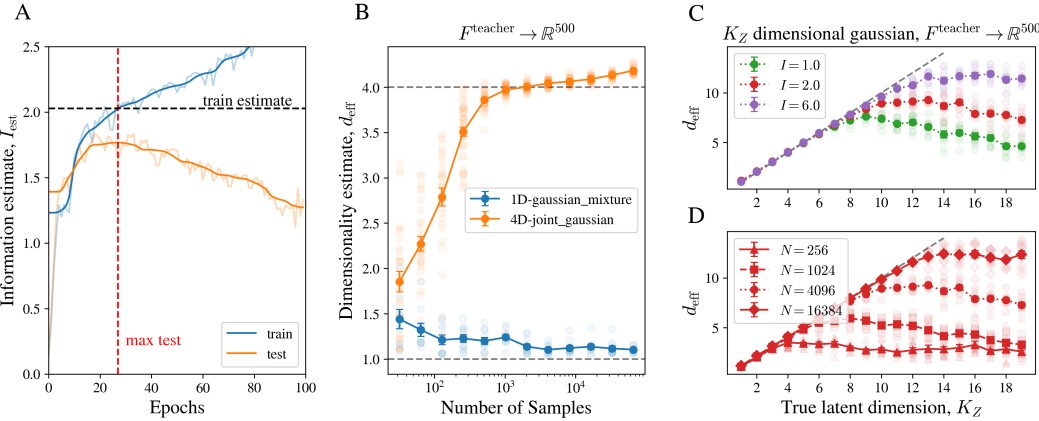

Figure 5: **Dimensionality estimation with finite data. (A)** Max-test, train-estimate protocol (Abdelaleem et al., 2025a) for a jointly Gaussian latent ($K_Z = 4$, $I = 2$ bits) observed through a teacher map into $\mathbb{R}^{500}$ with $N = 1024$ samples: select $t^* = \arg\max_t \widehat{I}_{\text{test}}(t)$ and report $\widehat{I}_{\text{train}}(t^*)$. Solid MI curves shown (and used to locate the maximum) are median-filtered over 20 epochs. **(B)** $d_{\text{eff}}$ from the trained encoders ($k_z = 64$) versus number of training samples for the two representative latent distributions (cross-covariance computed on the full training set). **(C,D)** $d_{\text{eff}}$ from a single model with $k_z = 64$ versus $K_Z$ for jointly Gaussian latents: varying total MI at fixed $N = 4096$ (C), and varying $N$ at fixed $I = 2$ bits (D). Usual notation for markers/error bars is used.

### 3.4 FINITE DATA

With a fixed finite dataset, variational MI bounds can overfit, so dimensionality estimation requires an explicit early-stopping rule. We use the *max-test, train-estimate* protocol of Abdelaleem et al. (2025a) (Fig. 5A): we select the training checkpoint $t^* = \arg\max_t \widehat{I}_{\text{test}}(t)$, but report $\widehat{I}_{\text{train}}(t^*)$ as the MI estimate. This choice is appropriate because our objective is estimating $I(X; Y)$ from the available sample, not optimizing predictive generalization, and it empirically reduces bias from overfitting. The same checkpoint yields encoder representations used for the participation-ratio estimate $d_{\text{eff}}$.

Figure 5B shows that $d_{\text{eff}}$ converges to the correct value for both representative latent distributions with $\sim 10^3$ samples. Figures 5C,D show $d_{\text{eff}}$ versus the true latent dimension $K_Z$ for jointly Gaussian latents across MI levels and dataset sizes. Our task-relevant dimensionality estimator remains accurate over a broad range of $K_Z$, failing only when the sample size is insufficient to resolve the latent signal. Taken together, these results show that latent dimensionality can be inferred from finite datasets using sufficiently expressive encoders, provided overfitting of MI estimates is controlled.

Our complete task-relevant dimensionality estimation protocol for finite data is:

---

**Task-relevant dimensionality estimation (finite data).** Given paired observations $(X, Y)$:

1. Choose $k_z$ larger than the expected task-relevant dimension.

2. Train an MI estimator with the hybrid critic using the max-test, train-estimate heuristic.

3. Compute the cross-covariance of the learned encoder representations.

4. Estimate dimensionality via the participation ratio $d_{\text{eff}}$ of the singular-value spectrum.

---

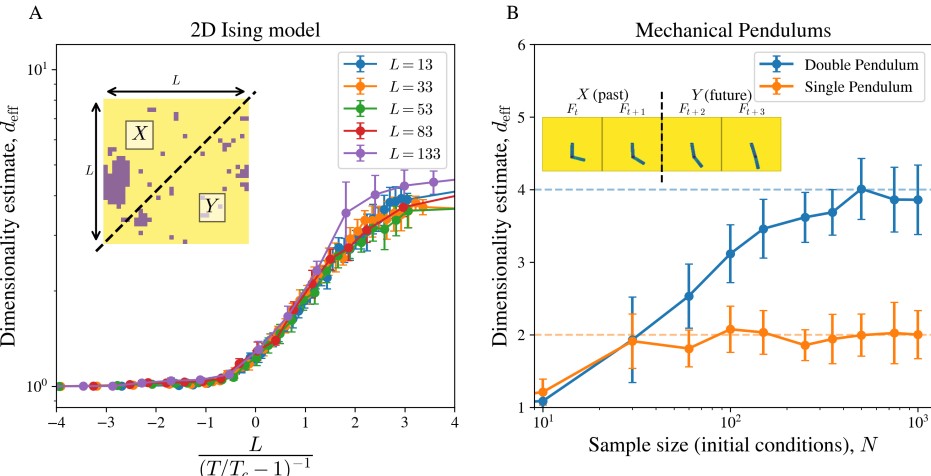

Figure 6: **Dimensionality estimation on physical datasets.** We apply the same protocol as in the synthetic benchmarks to (A) 2D nearest-neighbor Ising configurations ($J = 1$) and (B) single/double pendulum videos from Chen et al. (2022); error bars are standard deviations over 10 trials each. **(A)** $10^4$ MCMC samples on an $L \times L$ lattice; paired views $(X, Y)$ are formed by a spatial split of each configuration (inset). The measured $d_{\text{eff}}$ exhibits finite-size scaling and collapses across $L$ when plotted against the standard scaling variable $L/(T/T_c - 1)^{-1}$ with $\nu = 1$, with the collapse centered at $T_c$. **(B)** Paired views are constructed from the temporal structure of each video by stacking past and future frames (inset). With sufficient samples, $d_{\text{eff}}$ recovers the expected phase-space dimensions for the single (2) and double (4) pendulum.

## 3.5 Estimating task-relevant dimensionality of physics datasets

We now apply our dimensionality estimation protocol to more realistic datasets (simulated and experimental) whose latent structure is constrained by well-understood physics. In each case, we use the same estimator, training protocol, and network architectures as in the synthetic benchmarks, and test whether the inferred dimensionality reflects meaningful physical structure.

### 3.5.1 Ising model

Machine learning has been used to detect critical behavior in statistical mechanics with varying degrees of success (Carrasquilla & Melko, 2017; Giannetti et al., 2019; Carleo et al., 2019). Here we show that our dimensionality estimator identifies the phase transition and recovers the expected scaling in the 2D Ising model. In this system, near criticality, correlated domains grow and the correlation length diverges as $\xi \sim |T - T_c|^{-\nu}$, with $T_c = 2/\ln(1 + \sqrt{2}) \approx 2.269$ and $\nu = 1$ (Goldenfeld, 2018). The typical number of effectively independent domains spanning the shared interface scales as $\sim (L/\xi)$, suggesting that $d_{\text{eff}}$ should follow a finite-size scaling form and collapse across system sizes when plotted against the scaling variable $L/\xi \sim L/|T/T_c - 1|^{-\nu}$, with $\nu = 1$.

We simulate the Ising model on a periodic $L \times L$ lattice with $L \in [13, 133]$ (see App. F.1). To construct paired views from a single configuration, we use the view-splitting procedure of App. C.1, taking the upper-left half of spins as $X$ and the lower-right half as $Y$, and apply the same estimator and training protocol as in the synthetic benchmarks. The recovered effective dimensionality $d_{\text{eff}}$ tracks the growth of correlated domains (Fig. 15) and exhibits a clear scaling collapse across system sizes (Fig. 6A). The inferred dimensionality is therefore not an artifact of architecture or tuning, but reflects physically meaningful collective structure in the underlying system.

### 3.5.2 Pendulum dynamics

We next consider video recordings of simple mechanical systems and ask whether our estimator can recover the number of underlying degrees of freedom directly from raw pixels. Here $X$ is the delayed embedding (Takens, 2006) of two sequential "past" video frames, and $Y$ is the same of two future frames, so that the task-relevant dimensionality measures the number of degrees of freedom needed to predict future from the past. We analyze movies of a single pendulum (two degrees of freedom) and a chaotic double pendulum (four degrees of freedom), from Chen et al. (2022) (see App. F.2). We then apply our standard pipeline to infer the dimensionality of the underlying dynamics.

For this system, autoencoder based approached for dimensionality estimation have proven to be brittle (Chen et al., 2022). In contrast, our approach reliably infers the expected phase-space dimensions for both systems (Fig. 6B) from as few as a hundred samples, even in the chaotic pendulum.

## 4 Discussion

We proposed a method to measure *task-relevant* dimensionality: the minimal latent-state dimension needed to preserve the information required for a specified prediction problem. In our setting, relevance is defined by shared structure between paired views, quantified by MI. Thus "dimension" depends on the task and on how views are constructed (past–future or spatial splits), and is not a vague, scale-dependent intrinsic property of the raw observations.

Standard neural MI estimators are not automatically usable as dimensionality estimators. For example, separable/bilinear critics can systematically overestimate the dimension even in simple cases. Our *hybrid* critic addresses this by decoupling the interpretable bottleneck size (data geometry) from critic expressivity (architectural constraints). We also introduced a one-shot estimator based on the participation ratio of the cross-covariance spectrum in embedding space, combined with an early-stopping protocol that controls overfitting of variational MI bounds. This avoids sweeping $k_z$ and yields a stable estimate once the model is sufficiently overparameterized.

Limitations remain: the estimate depends on the success of the underlying MI estimation, on view construction, on the choice of embedding-network architecture, and on having enough samples to resolve latent structure. It will therefore be important to formalize conditions under which it succeeds or fails beyond the settings explored here. Nonetheless, unlike neighbor-scaling intrinsic-dimension estimators, our approach does not degrade quickly under observation noise, making it particularly useful for analyzing the latent representation geometry of noisy experimental datasets. The physics case studies support this: the resulting $d_{\text{eff}}$ exhibits finite-size scaling near the 2D Ising critical point and recovers phase-space dimensions from raw pendulum videos.

### Acknowledgments

We thank Sean Ridout for providing MCMC code to generate Ising spin configurations and for critical discussions. We thank K. Michael Martini for many discussions over the years. PG was funded, in part, by the Tarbutton Interdisciplinary Postdoctoral Fellowship at Emory College of Arts and Sciences. PG and IN were funded, in part, by the Simons Foundation Investigator grant to IN. EA and AS were supported, in part, by NIH Grant No. RF1-MH130413 and Brain and Behavior Research Foundation Young Investigator Grant 30885. We acknowledge support of our work through the use of the HyPER C3 cluster of Emory University's AI.Humanity Initiative.

### 4.1 Code Availability

The code to implement the general dimensionality estimation protocol, along with the synthetic dataset generation and real dataset analysis is available at https://github.com/paarthgulati/dim_est.

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

APPENDIX: TABLE OF CONTENTS

# A  THEORETICAL FRAMEWORK

## A.1  VARIATIONAL OBJECTIVES AND CRITIC CRCHITECTURES

Our approach is based on variational estimation of mutual information (MI) between two random variables $X$ and $Y$ using contrastive objectives and neural critics. We briefly summarize the relevant objective and critic parameterizations used throughout the paper, and fix notation for the theoretical analysis that follows.

**Variational MI estimators**  Mutual information can be written as a Kullback–Leibler divergence,

$$I(X;Y) = D_{\mathrm{KL}}(p(x,y)\|p(x)p(y)) = \mathbb{E}_{p(x)}\left[D_{KL}(p(y|x)\|p(y))\right]. \tag{5}$$

The Donsker-Varadhan (Donsker & Varadhan, 1983) representation provides a variational lower bound on the KL divergence between two distributions $P$ and $Q$:

$$D_{KL}(P\|Q) \geq \sup_T \left(\mathbb{E}_P[T] - \log \mathbb{E}_Q[e^T]\right), \tag{6}$$

where the supremum is taken over all measurable functions $T : \Omega \to \mathbb{R}$ on the common support, $\Omega$ of distributions $P$ and $Q$. Applying this to the various formulations of MI transforms the hard problem of calculating MI from estimating probabilities to optimizing over functions $T$. This is the basis of many MI estimation techniques, old and new (Nguyen et al., 2010; Belghazi et al., 2018; Poole et al., 2019; Song & Ermon, 2019).

Most neural-network-based MI estimation methods can be, directly or otherwise, derived from this formalism, with the most prominent estimators being MINE (Belghazi et al., 2018), SMILE (Song & Ermon, 2019) and InfoNCE (van den Oord et al., 2018). We do not aim to provide a comprehensive review of these estimators in our work, with better reviews available elsewhere (Poole et al., 2019). We focus on the symmetrized-InfoNCE estimator, which in addition to being a variational MI estimator, has been widely used in the larger representation learning community under the guise of symmetrized contrastive losses or SimCLR objectives (Chen et al., 2020; Radford et al., 2021).

We stress, however, that our approach to dimensionality estimation is not dependent on the choice of estimator and works perfectly well with other estimators mentioned above as they all admit an optimal critic (or family of critics) very similar to the symmetrized-InfoNCE estimator. We have conducted experiments with other estimators including SMILE and MINE, which in general do not suffer from log(batch size) bound of InfoNCE but suffer from larger variance with finite datasets (Belghazi et al., 2018; Song & Ermon, 2019). For simplicity of discussion, we present results throughout with the symmetrized-InfoNCE estimator.

To derive the **symmetrized-InfoNCE** objective, we apply the DV representation to the conditional form in Eq. (5), specifically to $D_{\mathrm{KL}}(p(y|x)\|p(y))$. We obtain:

$$I(X;Y) \geq \sup_T \left( \mathbb{E}_{p(x,y)}[T(x,y)] - \mathbb{E}_{p(x)}\left[\log \mathbb{E}_{p(y)}[e^{T(x,y)}]\right] \right). \tag{7}$$

Similarly, applying this DV representation to $D_{\mathrm{KL}}(p(x|y)\|p(x))$, we obtain:

$$I(X;Y) \geq \sup_{T'} \left( \mathbb{E}_{p(x,y)}[T'(x,y)] - \mathbb{E}_{p(y)}\left[\log \mathbb{E}_{p(x)}[e^{T'(x,y)}]\right] \right), \tag{8}$$

where $T'$ also independently consists of all measurable functions from the common support of $X$ and $Y$.

Of course, this implies

$$
\begin{aligned}
I(X;Y) \geq \frac{1}{2}\Big\{ &\sup_T \left( \mathbb{E}_{p(x,y)}[T(x,y)] - \mathbb{E}_{p(x)}\left[\log \mathbb{E}_{p(y)}[e^{T(x,y)}]\right] \right) \\
&+ \sup_{T'} \left( \mathbb{E}_{p(x,y)}[T'(x,y)] - \mathbb{E}_{p(y)}\left[\log \mathbb{E}_{p(x)}[e^{T'(x,y)}]\right] \right) \Big\} \\
\geq \sup_T &\left( \mathbb{E}_{p(x,y)}[T(x,y)] - \frac{1}{2}\left( \mathbb{E}_{p(x)}\left[\log \mathbb{E}_{p(y)}[e^{T(x,y)}]\right] + \mathbb{E}_{p(y)}\left[\log \mathbb{E}_{p(x)}[e^{T(x,y)}]\right] \right) \right) \\
:= &\mathcal{I}_{\text{symm-NCE}}(T)
\end{aligned}
\tag{9}
$$

From above, it is clear that $\mathcal{I}_{\text{symm-NCE}}(T)$ is bounded above by true MI. To show that it serves as a variational estimator, it suffices to show that there exists an optimal critic $T^*$ such that $\mathcal{I}_{\text{symm-NCE}}(T^*) = I$. This can be easily verified for $T^* = \log \dfrac{p(x,y)}{p(x)p(y)} + \mathrm{c}$. This is indeed the same optimal critic as for other variational estimators such as MINE (Poole et al., 2019).

In practice, the expectations are replaced by using contrastive sampling within a batch. For a batch $\{(x_i, y_i)\}_{i=1}^N$, we treat $y_i$ as the positive sample for $x_i$ and the other $N-1$ samples $\{y_j\}_{j \neq i}$ as negative samples from the marginal $p(y)$ (van den Oord et al., 2018). This yields the estimator:

$$I_{\text{symm-NCE}}(X,Y) := \frac{1}{2N}\left( \sum_{i=1}^N \log \frac{e^{T(x_i,y_i)}}{\frac{1}{N}\sum_{j=1}^N e^{T(x_i,y_j)}} + \sum_{j=1}^N \log \frac{e^{T(x_j,y_j)}}{\frac{1}{N}\sum_{i=1}^N e^{T(x_i,y_j)}} \right). \tag{10}$$

This estimator is low-variance but bounded above by $\log N$.

**Critic architectures** The MI estimators turn the MI estimation problem into an optimization problem of the variational bound, but the structure of the learned representation is controlled by the design of the critic function $T(x, y)$.

The critic determines how correlations between $X$ and $Y$ are represented. While sufficiently expressive critics can approximate the optimal solution of the variational problem by representing the optimal critic $T^*$, their internal structure governs whether correlations are able to be decomposed into low-dimensional degrees of freedom. This distinction plays a central role in our ability to extract an effective dimensionality from the representation optimized for MI estimation.

A common choice in neural MI estimation is a *concatenated* or joint critic (Belghazi et al., 2018), in which $X$ and $Y$ are combined by a generic function,

$$T_{\text{concat}}(x, y) = T_\theta([x, y]), \tag{11}$$

where $T_\theta$ is typically a MLP. This parameterization places minimal structural constraints on how correlations are represented. As a result, dependencies between $X$ and $Y$ can be distributed across the full joint representation, with no explicit notion of ordered degrees of freedom. While these critics are well suited for MI estimation, they do not expose a notion of intrinsic or effective dimensionality, and they are often more expensive to train, compared to the separable critic for example.

In contrast, another common design choice is *separable* critic architectures (van den Oord et al., 2018), which decompose the interaction between $X$ and $Y$ into a sum of factorized terms,

$$T_{\text{sep}}(x, y) = g^X(x) \cdot g^Y(y) = \sum_{k=1}^{k_z} g_k^X(x) g_k^Y(y). \tag{12}$$

Here, the critic represents a scalar product between embeddings of the two datasets via the *encoders* $g^X : \mathcal{X} \to \mathbb{R}^{k_z}$ and $g^Y : \mathcal{Y} \to \mathbb{R}^{k_z}$. This form explicitly enforces a pairing structure between learned features of the two variables and decomposed into (at most) $k_z$ modes.

Separable and concatenated critics impose qualitatively different constraints on how correlations are represented, but neither is naturally useful for measuring the latent dimensionality: separable critics restrict interactions to a bilinear form at a prescribed embedding dimension[2], while concatenated critics are flexible but can entangle dependencies across coordinates, obscuring their organization.

In this work, we bridge the gap and motivate a ***hybrid*** architecture that can force accurate low-dimensional, latent space representations and then optimize over a flexible concatenated head to represent the optimal critic in the learned latent space. As we demonstrate throughout the paper, this separation allows the encoders to faithfully represent the latent degrees of freedom, while the concatenated head optimizes the variational objective without imposing additional structural constraints. The effective dimensionality is therefore reflected in the learned embeddings themselves:

$$T_{\text{hybrid}}(x, y) = T_\theta([g^X(x), g^Y(y)]). \tag{13}$$

In principle, the choice of critic architecture can be entirely decoupled from the choice of the variational estimator (i.e., objective). In this paper, we show that the *hybrid* architecture, coupled with the symmetrized-InfoNCE, not only can accurately estimate MI, sometimes outperforming existing estimators in the number of samples required (Fig. 14), but also reveals the correct latent space dimensionality across a wide variety of latent spaces and distributions.

## A.2 Exactly solvable case: Jointly Gaussian variables

In the case of jointly Gaussian variables, the optimal critic $T^*$ is known analytically. By examining the functional form of the optimal critic in this simple setting, we can validate

---

[2]The relevance of latent dimensionality for NN-based MI estimators is becoming more popular (see, e.g, Gowri et al. (2024)), but such work has relied on using existing separable architectures, which, as we have shown comes with stark limitations and inflated dimensionality estimation.

our numerical results, demonstrate the limitations of a separable architecture and highlight the utility of the *hybrid* design.

### A.2.1 DERIVATION OF THE OPTIMAL GAUSSIAN CRITIC

This derivation largely follows the pedagogical introduction in Abdelaleem et al. (2025a). Consider $X \in \mathbb{R}^{K_X}$ and $Y \in \mathbb{R}^{K_Y}$ distributed as a zero-mean jointly Gaussian variable $Z = [X, Y]^\top$:

$$Z \sim \mathcal{N}(0, \Sigma), \quad \Sigma = \begin{bmatrix} \Sigma_{XX} & \Sigma_{XY} \\ \Sigma_{YX} & \Sigma_{YY} \end{bmatrix}. \tag{14}$$

The optimal critic (for the population-level variational objective underlying the symmetrized-InfoNCE estimator), i.e., $\mathcal{I}_{\text{symm-NCE}}(T^*) = I(X; Y)$, is given by:

$$T^*(x, y) = \log \frac{p(x, y)}{p(x)p(y)} + C. \tag{15}$$

Substituting the probability density functions for the multivariate Gaussian distribution,

$$p(x, y) \propto \exp\left(-\frac{1}{2}\begin{bmatrix} x \\ y \end{bmatrix}^\top \Sigma^{-1} \begin{bmatrix} x \\ y \end{bmatrix}\right), \quad p(x)p(y) \sim \exp\left(-\frac{1}{2}x^\top \Sigma_{XX}^{-1} x - \frac{1}{2}y^\top \Sigma_{YY}^{-1} y\right), \tag{16}$$

yields a quadratic form for the optimal critic

$$T^*(x, y) = \frac{1}{2}\left[x^\top \Sigma_{XX}^{-1} x + y^\top \Sigma_{YY}^{-1} y - \begin{bmatrix} x \\ y \end{bmatrix}^\top \Sigma^{-1} \begin{bmatrix} x \\ y \end{bmatrix}\right] + C. \tag{17}$$

This explicitly shows that the optimal critic for jointly Gaussian data is a quadratic function of the inputs. To analyze this, it is useful to transform to a canonical coordinate system.

To do so, define the whitened (i.e., normalized) cross-covariance matrix, $\mathcal{K} = \Sigma_{XX}^{-1/2}\Sigma_{XY}\Sigma_{YY}^{-1/2}$ and then perform its singular value decomposition, $\mathcal{K} = U\Lambda V^\top$, where $\Lambda = \text{diag}(\rho_1, \ldots, \rho_r)$, with $r = \text{rank}\,\mathcal{K}$. We can define the canonical variables $u$ and $v$ via linear projections

$$u = U^\top \Sigma_{XX}^{-1/2} x, \quad v = V^\top \Sigma_{YY}^{-1/2} y. \tag{18}$$

In this coordinate system, the joint covariance matrix $\Sigma$ simplifies into blocks for each canonical pair $(u_i, v_i)$ with the inverse covariance matrix $\Sigma^{-1}$ for the $i$-th pair given by:

$$(\Sigma^{(i)})^{-1} = \frac{1}{1 - \rho_i^2}\begin{bmatrix} 1 & -\rho_i \\ -\rho_i & 1 \end{bmatrix}. \tag{19}$$

Using this, the optimal critic can be written as a sum over independent canonical coordinates, decomposed into bilinear and quadratic terms:

$$T^*(x, y) = \sum_{i=1}^{r}\left[\frac{\rho_i}{1 - \rho_i^2}u_i v_i - \frac{1}{2}\frac{\rho_i^2}{1 - \rho_i^2}(u_i^2 + v_i^2)\right]. \tag{20}$$

The first interaction term is purely *bilinear* in $(u_i v_i)$, while the *self-normalization* terms are quadratic in $u_i$ and $v_i$. This derivation effectively recovers the standard Canonical Correlation Analysis (CCA) objective (Kullback, 1959; Gelfand, 1959). Below, we use this expression to design a quadratic critic architecture (*separable-augmented*) for the jointly Gaussian case, demonstrating how it indeed can be used to infer the dimensionality of the latent distribution, as well as explore the role of embedding dimension in separable critics.

A.2.2  CHOICE OF ARCHITECTURE AND OPTIMAL EMBEDDINGS

Given the optimal critic, Eq. (20), it is clear that a $K$ dimensional jointly Gaussian distribution is not in the class of separable critics with encodings in $k_z = K$ dimensions (here and throughout this section we simplify notation with $K$ in place of $K_Z$ for the latent dimensionality). In particular, the "self-normalization" terms cannot be represented as a dot product in $K$-dimensional space.

However, this is typically not a limitation for MI estimation using separable architectures, where the encoders are non-linear and typically embed in a much larger space than the true latent structures. To see that a bilinear representation of an optimal critic is possible with non-linear encoders in $K + 2$ dimensions, we can directly check that $T(x, y) = g^X \cdot g^Y = T^*(x, y)$ for the following encoding:

$$
\begin{aligned}
g^X(x) &= \left( \sqrt{\frac{\rho_1}{1 - \rho_1^2}} u_1, \cdots, \sqrt{\frac{\rho_K}{1 - \rho_K^2}} u_K, \sum_i^K -\frac{\rho_i^2}{2(1 - \rho_i^2)} u_i^2, 1 \right) \in \mathbb{R}^{K+2}, \\
g^Y(y) &= \left( \sqrt{\frac{\rho_1}{1 - \rho_1^2}} v_1, \cdots, \sqrt{\frac{\rho_K}{1 - \rho_K^2}} v_K, 1, \sum_i^K -\frac{\rho_i^2}{2(1 - \rho_i^2)} v_i^2 \right) \in \mathbb{R}^{K+2}.
\end{aligned}
\tag{21}
$$

This is one of many degenerate solutions that can represent the optimal critic via a dot product in $K + 2$ dimensions. The scalar product in the first $K$ dimensions captures the interaction terms in Eq. (20), and the two extra dimensions individually capture the normalization terms for $X$ and $Y$.

Note that this is only for estimators (MINE, SMILE, symmetrized-InfoNCE) where the optimal critic corresponds to $T^* = \log \left( p(x, y)/p(x)p(y) \right) + C$. With nonsymmetrized InfoNCE, the optimal critic is given by $T^* = \log p(y|x) + \tilde{c}(y) = \log \left( p(x, y)/p(x)p(y) \right) + c(y)$ (Poole et al., 2019), with the functional degeneracy represented in $c(y)$. In this case, with the separable architecture the optimal critic family can clearly be encoded in $K + 1$ dimensions, by, e.g., the first $K + 1$ dimensions of the encoders given in Eq. (21) and discarding the normalization terms for $y$.

To validate that the correct MI estimation (with a known latent distribution) corresponds to the critic learning the correct latent representation Eq. (20), we construct an extended quadratic family of critics that can be learned while training, and we explore whether such an architecture allows for the optimal critic to be learned in $K$ dimensions exactly. Using encoders $g^X(x)$ and $g^Y(y)$, we define:

$$
T_{\text{separable-augmented}}(x, y) = (g^X)^\top g^Y + (g^X)^\top \gamma^X g^X + (g^Y)^\top \gamma^Y g^Y
\tag{22}
$$

where, $\gamma^X, \gamma^Y$ are $(k_z \times k_z)$ weight matrices composed of trainable parameters. With correctly learnt encoders and $\gamma^X, \gamma^Y$ it is obvious that the optimal critic, Eq. (20), can be represented by the *separable-augmented* critics in $k_z = K$ dimensions. This is indeed what we find, as shown in Fig. 7.

A curious observation in these experiments is that even with the *separable* critics and symmetrized-InfoNCE estimator such that the optimal critic $T^*(x, y) = \log \left( p(x, y)/p(x)p(y) \right)$, we empirically find that $K + 1$ dimensions suffice to learn the true MI (and not $K + 2$ as we argued above would, at most, suffice). There is no obvious solution in $K + 1$ dimensions that would actually be sufficient to represent this optimal critic. Visualizing the learnt encoders (with $k_z = K + 1$) in our experiments with the joint Gaussian distributions, we observe spherical embeddings (not shown). This leads us to consider an interesting ansatz: encoding on hypersphere $S^K$, embedded in $\mathbb{R}^{K+1}$, can lead to accurate MI for a $K$-dimensional joint Gaussian with separable critics.

To verify this, we first restrict to $K = 1$, i.e., 1-dimensional joint Gaussians, with embeddings:

$$
\begin{aligned}
g^X(x) &= \left( \alpha_1 \cos(u/\beta_1), \alpha_2 \sin(u/\beta_1) \right), \\
g^Y(y) &= \left( \alpha_3 \cos(v/\beta_2), \alpha_4 \sin(v/\beta_2) \right),
\end{aligned}
\tag{23}
$$

where, as before, $u, v$ are the canonical coordinates corresponding to $x, y$ respectively such that $\mathbb{E}[u] = \mathbb{E}[v] = 0$, $\mathbb{E}[u^2] = \mathbb{E}[v^2] = 1$. The correlation $\rho$ is defined as $\rho = \mathbb{E}[uv]$ and then

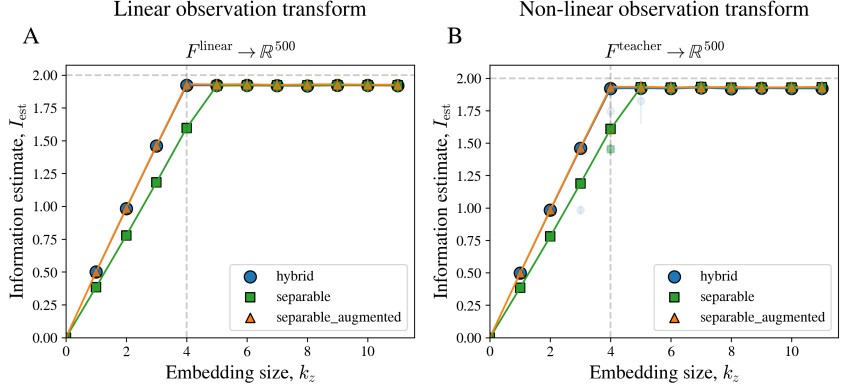

Figure 7: **Required embedding dimensionality for different critics.** The separable augmented critic, with the *symmetrized-InfoNCE* estimator also recover the true MI, in embedding dimension $k_z = K_Z$ for a joint Gaussian distribution. The figures supplements Fig. 2 in the main text, for the joint Gaussian distribution with $K_Z = 4$, passed through linear and non-linear teacher transforms to $\mathbb{R}^{500}$ to create the observed data. Individual runs shown by semi-transparent markers.

the true MI is $I = -\frac{1}{2}\log_2(1 - \rho^2)$. Here, $\alpha_i, \beta_i$ are parameters we will use to optimize the MI estimate from this embedding.

The separable critic is then given by

$$
\begin{aligned}
T(x,y) = g^X \cdot g^Y &= \alpha_1\alpha_3\cos(u/\beta_1)\cos(v/\beta_2) + \alpha_2\alpha_4\sin(u/\beta_1)\sin(v/\beta_2) \\
&= \lambda_+ \cos(au + bv) + \lambda_- \cos(au - bv),
\end{aligned}
\tag{24}
$$

with $\lambda_\pm = \frac{1}{2}(\alpha_1\alpha_3 \mp \alpha_2\alpha_4)$. Here, for tractability, we restrict ourselves to $\lambda_+ = 0$, which can only provide a lower bound on the estimated MI. As we will show, even with this restricted subspace, we can approximate the true MI very well. And this restriction corresponds to $\alpha_1 = \alpha_2, \alpha_3 = \alpha_4$ i.e. a circular embedding. So, we optimize the symmetrized-InfoNCE objective over $\lambda, a, b$ where

$$
T(x,y) = \lambda\cos(au - bv). \tag{25}
$$

With this restricted critic, to optimize the symmetrized-InfoNCE estimate over the critic means to optimize the bound:

$$
\mathcal{I}(a,b,\lambda;\rho) = \mathbb{E}[T] - \frac{1}{2}\mathbb{E}_X\log\mathbb{E}_Y e^T - \frac{1}{2}\mathbb{E}_Y\log\mathbb{E}_X e^T \tag{26}
$$

such that

$$
\mathcal{I}^{\text{circle}}(\rho) \equiv \sup_{a,b,\lambda} \mathcal{I}(a,b,\lambda;\rho). \tag{27}
$$

To compute this optimal estimate, $\mathcal{I}^{\text{circle}}$, we make use of two identities:

1. For $W \sim \mathcal{N}(\mu, \sigma^2)$

$$
\mathbb{E}[e^{inW}] = e^{in\mu}e^{-\frac{1}{2}n^2\sigma^2}. \tag{28}
$$

   Which directly implies, $\mathbb{E}[\cos(nW)] = \cos(n\mu)e^{-n^2\sigma^2/2}$.

2. To calculate the expectation over the exponential of a sinusoid $\mathbb{E}[e^{\lambda\cos(au-bv)}]$, we use the Jacobi expansion using Bessel functions and then compute the expectations as above with

$$
e^{z\cos\theta} = I_0(z) + 2\sum_{n=1}^{\infty} I_n(z)\cos(n\theta), \tag{29}
$$

   where $I_n$ are modified Bessel functions of the first kind.

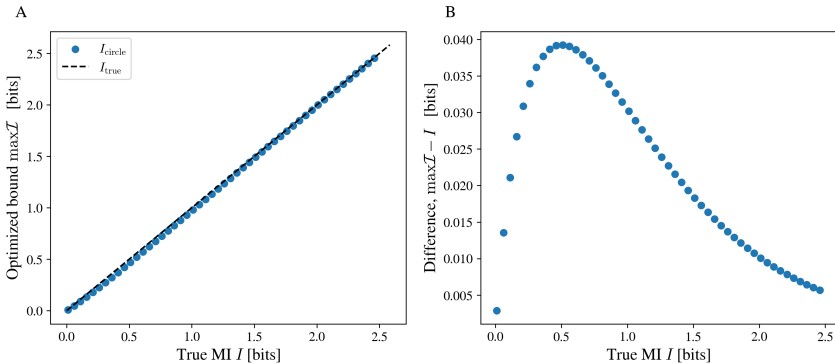

Figure 8: **Bilinear critic produces circular embeddings.** We estimate the variational bound for the circular embedding by numerically approximating $\mathcal{I}(\kappa, \lambda|\rho)$ Eq. (32) using quadratures and then optimizing over a grid of $\kappa, \lambda$ to get the variational bound $\mathcal{I}^{\mathrm{circle}}(\rho)$. As discussed in the text, we find that this estimator can approach the true MI, with a gap $< 0.04$ bits, showing how a two dimensional separable critic can approach the true MI for a one-dimensional joint Gaussian distribution.

With these identities, we can write down the various expectations needed for the estimator in Eq. (26). The first term is given by:

$$\mathbb{E}[T] = \lambda \exp\left[-\frac{1}{2}(a^2 + b^2 - 2ab\rho)\right] . \tag{30}$$

For the log terms, we cannot compute the expectations analytically, but we can reduce it to quadrature, as below, and then numerically optimize over $a, b$, and $\lambda$. This gives:

$$
\begin{aligned}
\mathbb{E}_Y e^T &= \mathbb{E}_Y\left[e^{\lambda \cos(au - bv)}\right] \\
&= \mathbb{E}_Y\left[I_0(\lambda) + 2\sum_{n=1}^{\infty} I_n(\lambda)\cos(n(au - bv))\right] \\
&= I_0(\lambda) + 2\sum_{n=1}^{\infty} I_n(\lambda)\cos(nau)e^{-n^2 b^2/2}.
\end{aligned}
\tag{31}
$$

And then we can compute $\mathbb{E}_X \log(\mathbb{E}_Y e^T)$ numerically.

From symmetry of $a \rightleftharpoons b$ in the full objective function, (given a unique optimizer) the optimum will lie at $a = b := \kappa$ and then both the row and column terms are equal, such that

$$\mathcal{I}(\kappa, \lambda; \rho) = \lambda e^{-\kappa^2(1-\rho)} - \mathbb{E}_Z\left[\log\left(I_0(\lambda) + 2\sum_{n=1}^{\infty} I_n(\lambda)\cos(n\kappa z)e^{-n^2\kappa^2/2}\right)\right], \tag{32}$$

where $z \in Z \sim \mathcal{N}(0, 1)$, $I_n$ are the modified Bessel functions of the first kind, and $I^{\mathrm{circle}}(\rho) \geq \max_{\kappa, \lambda} \mathcal{I}(\kappa, \lambda; \rho)$.

We approximate this expectation over Gaussian $z$ numerically using Gauss-Hermite polynomials, and as shown in Fig. 8, we find that $\max|I_{\mathrm{circle}} - I| \approx 0.04$ bits, with the peak at $I \approx 0.5$ bits or equivalently $\rho = \sqrt{1 - 2^{-2I}} \approx 0.7$. Such a gap would be well within the fluctuations (due to finite data/batch sizes) in our experiments with the separable critics and, therefore, would register as a saturation of the MI estimate for $k_z = K + 1$.

Note, we have not shown this to be true for $K > 1$. For the joint Gaussian, with equal correlations in $K$ directions, as has been considered throughout this paper, a natural extension of this circular embedding is to embed on an $S^K$-sphere in $\mathbb{R}^{K+1}$, i.e., with angles $\theta_i = \kappa_i u_i$ such that the encoder $g^X$ has components $g_1 = \lambda \cos\theta_1$, $g_2 = \lambda \sin\theta_1 \cos\theta_2, \cdots,$

$g_K = \lambda \sin\theta_{K-1} \cdots \sin\theta_1 \cos\theta_K$, $g_{K+1} = \lambda \sin\theta_K \cdots \sin\theta_1$, and analogously for $g^Y$. With the same symmetry constraints as in the 1-d case with $\kappa \equiv \kappa_i^X = \kappa_i^Y$, and using a more general expansion of the $d$-dimensional sinusoidal products, one can then consider a more general expansion of the exponential expectations.

While we do not pursue this calculation further here, and consider this beyond the scope/relevance of this work, our numerical experiments with separable critics and the symmetrized-InfoNCE estimator strongly suggest that such hyperspherical embeddings are sufficient to achieve near-saturation of MI bound with embedding dimension $k_z = K + 1$. This observation is consistent with general empirical findings in representation learning, where contrastive losses (which as we showed earlier, are equivalent to MI optimization) and separable architectures (e.g., SimCLR) tend to produce approximately uniform and aligned embeddings on the hypersphere in sufficiently high-dimensional latent spaces (Wang & Isola, 2020). In sufficiently high-dimensional latent spaces, a wide class of non-pathological distributions exhibit approximately Gaussian statistics along most directions, essentially due to aggregation of independent factors (Diaconis & Freedman, 1984), and, hence, the optimal embedding which maximizes the variational MI objectives, using separable architectures, would result in hyperspherical embeddings.

## B    DESIGN CHOICES: ESTIMATION AND STOPPING PROTOCOLS AND EFFECTIVE DIMENSIONALITY MEASURES

### B.1    MAX-TEST / TRAIN-ESTIMATE PROTOCOL

A fundamental challenge that is often overlooked in neural-network-based MI estimators when training on finite datasets is determining when to stop training. Unlike supervised learning, where a test loss plateau indicates convergence, variational MI lower bounds can grow indefinitely as the critic overfits to finite-sample artifacts. For example, InfoNCE can keep growing until saturation at $\log N$ while SMILE can keep growing past that point.

To resolve this, we follow Abdelaleem et al. (2025a) and employ a **Max-Test / Train-Estimate** protocol. We split the available data into a training set ($\mathcal{D}_{\text{train}}$) and a test set ($\mathcal{D}_{\text{test}}$). At each training epoch $t$, we evaluate the estimator on both sets:

$$\hat{I}_{\text{train}}^{(t)} = \mathcal{L}_{\text{EST}}(\mathcal{D}_{\text{train}}, \theta_t), \tag{33}$$

$$\hat{I}_{\text{test}}^{(t)} = \mathcal{L}_{\text{EST}}(\mathcal{D}_{\text{test}}, \theta_t). \tag{34}$$

We select the optimal stopping point $t^*$ as the epoch that maximizes the test estimate: $t^* = \arg\max_t \hat{I}_{\text{test}}^{(t)}$. However, crucially, we report $\hat{I}_{\text{train}}^{(t^*)}$ as the final estimate, rather than the test value.[3]

### B.2    EFFECTIVE DIMENSIONALITY

To quantify the effective dimensionality of the learned representations without performing an exhaustive sweep over embedding sizes $k_z$, we analyze the spectrum of the learned embeddings. We train a single *hybrid* estimator with an overparameterized bottleneck ($k_z \gg K_Z$).

Let $Z_X = g^X(X)$ and $Z_Y = g^Y(Y)$ be the batch of embeddings produced by the encoders. We compute the centered cross-covariance matrix:

$$C_{XY} = \frac{1}{N-1}(Z_X - \bar{Z}_X)^\top (Z_Y - \bar{Z}_Y). \tag{35}$$

---

[3]Several practical choices can reduce overhead. One can monitor a proxy for $\hat{I}_{\text{train}}$ by evaluating on a subset of the training data rather than the full set. One can also avoid computing $\hat{I}_{\text{train}}$ at every epoch: evaluate only the test estimate, save the checkpoint whenever $\hat{I}_{\text{test}}$ improves, and at the end compute $\hat{I}_{\text{train}}^{(t^*)}$ once on the full training set using the best checkpoint. Note that for dimensionality estimation knowing the MI value per se is not crucial (e.g., in settings like $I(g^X(Z); g^Y(Z))$), but it is still useful to track MI to detect estimator saturation (e.g., the $\log N$ ceiling for InfoNCE) and to decide whether to switch estimators, stop earlier, increase the batch size, etc. Finally, runs that fail to learn any nontrivial MI should be discarded.

We then compute the singular values $\{\sigma_i\}$ of $C_{XY}$. The effective dimensionality $d_{\text{eff}}$ is derived from this spectrum. The choice of metric for $d_{\text{eff}}$ involves a choice of how strongly small values in the spectrum are counted or what constitutes a gap in the spectrum, which resembles how do we decide that we see a saturation in $I$ vs $k_z$ curves in case of absence of a clear sharp knee in the curve.

In this paper, we define $d_{\text{eff}}$ via the Participation Ratio (PR) of the singular values:

$$d_{\text{eff}} = \frac{(\sum_i \sigma_i)^2}{\sum_i \sigma_i^2}. \tag{36}$$

This choice of metric corresponds to a choice of how to filter the learned spectrum into an effective scalar dimensionality estimate. This could be replaced with different choices: we can either threshold the (normalized) singular value spectrum to count the number of non-trivial modes, or define various continuous analogues to the participation ratio, e.g.:

1. PR based on Eigenvalues: Similar to Eq. (36), we can define:

$$d_{\text{eff}}^{\text{eig}} = \frac{(\sum_i \lambda_i)^2}{\sum_i \lambda_i^2} = \frac{(\sum_i \sigma_i^2)^2}{\sum_i \sigma_i^4}. \tag{37}$$

   Such definition kills smaller values of the spectrum faster, yielding a smaller estimate for $d_{\text{eff}}$.

2. PR based on Spectral Entropy: Defining dimensionality as the exponential of the spectral entropy, $e^{H(\tilde{\sigma})}$, where $\tilde{\sigma}_i = \sigma_i / \sum_i \sigma_i$ are normalized singular values:

$$d_{\text{eff}}^{\text{s−ent}} = \exp\{-\sum_i \tilde{\sigma}_i \log \tilde{\sigma}_i\}. \tag{38}$$

   Such a definition often inflates the dimensionality estimate by including smaller components, leading to estimates of $d_{\text{eff}} > K_{\text{true}}$ in noisy regimes.

3. Intermediate PR:

$$d_{\text{eff}}^{\alpha} = \frac{(\sum \sigma_i^{\alpha})^2}{\sum \sigma_i^{2\alpha}} \tag{39}$$

As illustrated in Fig. 9, various choices of such a metric relate to how to consider contributions from unequally dominant modes in the embeddings, and, in principle, should depend on the physical system under consideration. For concreteness, throughout this paper, we default to the PR of the singular values as our metric of choice as it provides a robust middle ground, but there is no a priori reason for such a choice. For spectra with equally dominant modes, all the choices would converge and provide very similar estimates of effective dimensionality.

## C    Dimensionality estimation with observation noise

### C.1    View splitting for intrinsic dimensionality estimation

In Sec. 3, we validated our estimator on synthetic datasets where distinct views $X$ and $Y$ were explicitly generated. However, in scientific applications (Sec. 3.5), we often possess a single high-dimensional dataset (e.g., a video or a spin configuration) and wish to infer its intrinsic dimensionality. Here, we provide the theoretical and empirical justification for applying our estimator in this "single-dataset" regime. We model the data as arising from a *Shared Latent Space* model:

$$X = F_X(Z) + \eta_X, \quad Y = F_Y(Z) + \eta_Y, \tag{40}$$

where $Z$ is the underlying latent variable (drawn from $p_Z$), $F_X, F_Y$ are (potentially identical) observation maps, and $\eta_X, \eta_Y$ are uncorrelated noise terms.

Crucially, the presence of observation noise is not a hindrance but a requirement for dimensionality estimation via MI. In the absence of noise ($\eta \to 0$), if $F$ is deterministic and

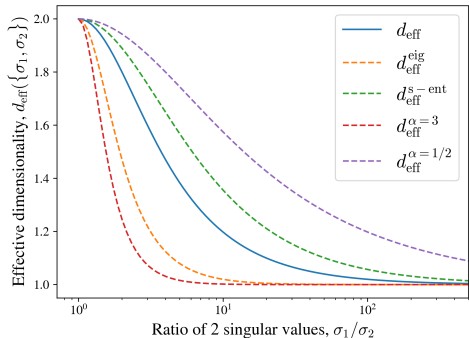

Figure 9: **Effective dimensionality based on the various metrics for a pair of singular values, $\sigma_1$ and $\sigma_2$, as a function of their ratio.** With both singular values equal, the effective dimensionality is 2. As they become unequal, the effective dimensionality becomes smaller, with the rate of decrease determined by the choice of the metric. Throughout this paper, we have used the participation ratio of the singular values (shown here with the solid line) of the cross-covariance of the encoders as the measure for effective dimensionality.

invertible, MI would diverge (or, more precisely, equal the entropy $H(Z)$, which is parameterization dependent), making it difficult to define a saturation point relative to the embedding dimension. With uncorrelated noise, the information bottleneck is determined by the shared signal $Z$, and one expects the estimated MI to saturate when the embedding dimension $k_z$ matches $\dim(Z)$.

We validate this on two challenging latent topologies: a hypersphere ($K_Z = 3$) and a Swiss roll ($K_Z = 2$). Two high-dimensional noisy views are generated as in Eq. (40) (see App. E.1 for details). As shown in Fig. 10, even when the observation maps project these manifolds into $K = 500$ dimensions, our estimator identifies the saturation point at $k_z = K_Z$.

This justifies the view-splitting strategy employed in Sec. 3.5. While for the pendulum, taking past as $X$ and future as $Y$ is natural, for the Ising model, we can instead partition each configuration into two spatial domains, yielding views that are conditionally independent given the latent state $Z$. Our estimator then recovers the dimensionality of this shared latent structure.

## C.2    COMPARING WITH INTRINSIC DIMENSIONALITY ESTIMATORS

We compare our view-splitting, hybrid-critic MI dimensionality protocol with standard intrinsic dimensionality (ID) estimators: the Levina–Bickel maximum-likelihood estimator (MLE) (Levina & Bickel, 2004) (with an adaptive neighborhood size) and the Two–Nearest-Neighbor estimator (Two-NN) (Facco et al., 2017), implemented using the `scikit-dimension` library (Bac et al., 2021). In Fig. 11 we use the same synthetic construction as in Sec. 3.2: data are generated from a jointly Gaussian latent, passed through the teacher network, and corrupted by additive observation noise. The ID estimators are applied separately to the $X$ and $Y$ point clouds, whereas our task-relevant dimensionality estimate is computed from the paired dataset $(X, Y)$.

In this setting, all estimators recover the latent dimensionality in the absence of added observation noise ($\eta = 0$). With observation noise, however, the geometric ID estimators degrade sharply, tending towards the ambient observation dimension. In contrast, our protocol—computing $d_{\text{eff}}$ via the participation ratio of the learned latent cross-covariance spectrum—remains robust and continues to recover the latent dimensionality over the noise levels tested.

While this behavior is expected (ID estimators cannot separate latent structure from observation noise, whereas the paired-view formulation targets shared signal), it highlights the practical value of *task-relevant* dimensionality: for high-dimensional scientific data mired in

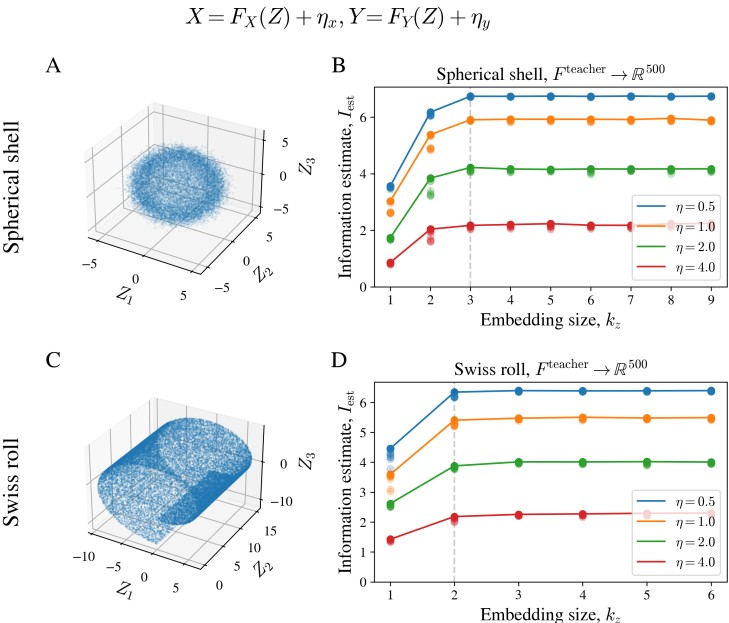

$$X = F_X(Z) + \eta_x, Y = F_Y(Z) + \eta_y$$

Figure 10: **Estimating intrinsic dimensionality by view splitting.** We infer the dimensionality of a shared latent variable $Z$ from two noisy views $X$ and $Y$. **(A,C)** Latent manifolds: (A) hypersphere ($K_Z = 3$) and (C) Swiss roll ($K_Z = 2$). **(B,D)** Estimated MI versus $k_z$. With observation noise $\eta > 0$, the estimate saturates at the true latent dimensionality $K_Z$, recovering the degrees of freedom of the shared signal. Error bars are standard deviations over 10 trials (semi-transparent markers).

observational noise, the paired-view MI approach can remain informative in regimes where standard ID estimators do not.

Recent work has improved intrinsic dimensionality estimation in specific regimes, such as locally undersampled manifolds and high-dimensional binary datasets Erba et al. (2019); Acevedo et al. (2025). These methods can outperform the classic estimators used here as baselines in the regimes they target. However, they still estimate geometric intrinsic dimension which, in the absence of paired views, makes it difficult to disentangle observation noise from variance in the latent space.

# D   ADDITIONAL RESULTS

## D.1   NON-INVERTIBILITY OF THE GAUSSIAN MIXTURE

The Gaussian mixture distribution (see App. E.1 for the density), constructed to have multiple overlapping clusters, is characterized by the number of clusters $N_{\text{peaks}}$, the ring radius $\mu$ in $Z_X$–$Z_Y$ space, and the within-cluster correlation $\rho$. For $N_{\text{peaks}} > 1$ the joint distribution is multimodal, and the conditional structure is generically non-invertible. As shown in the main text (Fig. 2E,F) for our representative mixture ($N_{\text{peaks}} = 8$, $\mu = 2.0$, $\rho \approx 0.97$), dimensionality estimates based on MI saturation with a separable critic are inflated. We see now that the same conclusion holds for the participation-ratio method developed in this paper (Fig. 12).

We further conjecture that this inflation reflects an implicit decomposition of the latent space into patched within which the joint distribution is approximately Gaussian, so that a bilinear critic can represent the dependence locally. In this view, the minimal number of such regions (e.g., admitting approximate Gaussian copulas) sets the effective dimensionality required by the separable critic (Kulpa, 1999; Ghosh & Henderson, 2009). This interpretation is consistent with the dependence of the separable $d_{\text{eff}}$ on the number of mixture components

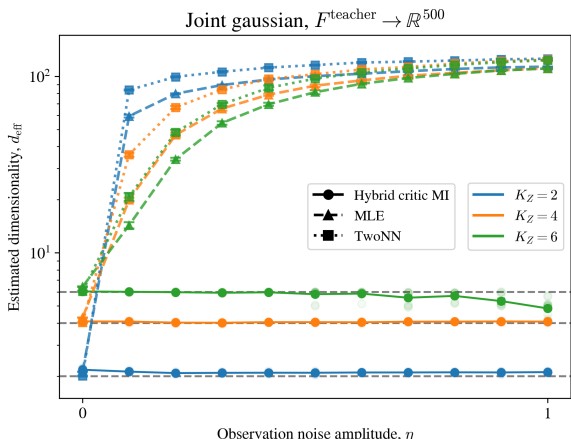

Figure 11: **Comparison of intrinsic dimensionality estimators with our paired-view MI protocol.** We use a finite dataset of size $N = 16{,}384$ generated from a $K_Z$-dimensional Gaussian latent, mapped through teacher transforms into $K = 500$ observed dimensions, with added uncorrelated observation noise. In the absence of noise, all methods recover the latent dimension. With observation noise ($\eta > 0$), MLE and Two-NN degrade and cannot decouple latent dimensionality from observation-space noise, while our hybrid-critic MI protocol remains robust over the range shown. For MLE and Two-NN, we report the average of the estimates obtained from applying the estimator to $X$ and to $Y$. All methods averaging over 10 independent trials; error bars are standard deviations.

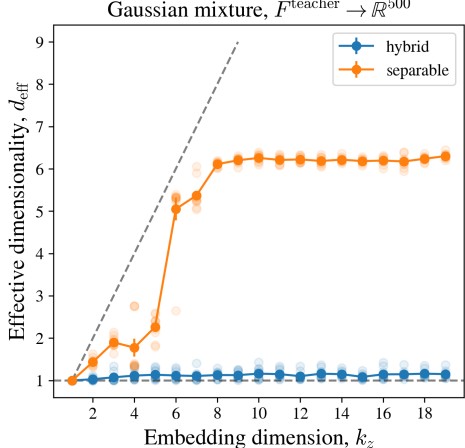

Figure 12: **Separable critics inflate dimensionality for multimodal latents.** For the representative Gaussian mixture ($N_{\text{peaks}} = 8$, $\mu = 2.0$, $I_p = 2.0$ bits), the separable critic yields an inflated effective dimensionality, $d_{\text{eff}} \approx 7$, whereas the hybrid critic recovers the latent dimensionality. We conjecture that the inflation reflects the need to represent a multimodal dependence structure using bilinear product terms. Averages over 10 independent trials; error bars are standard deviations.

(Fig. 13). For example, for our parameters, the estimated $d_{\text{eff}}$ decreases at $N_{\text{peaks}} = 4$ as clusters collapse pairwise due to the ring symmetry (see representative samples in Fig. 13), and similarly at $N_{\text{peaks}} = 6$ (not shown). Under this interpretation, separable critics may also provide an empirical probe of multimodality in low-dimensional latent distributions; we defer a quantitative analysis to future work.

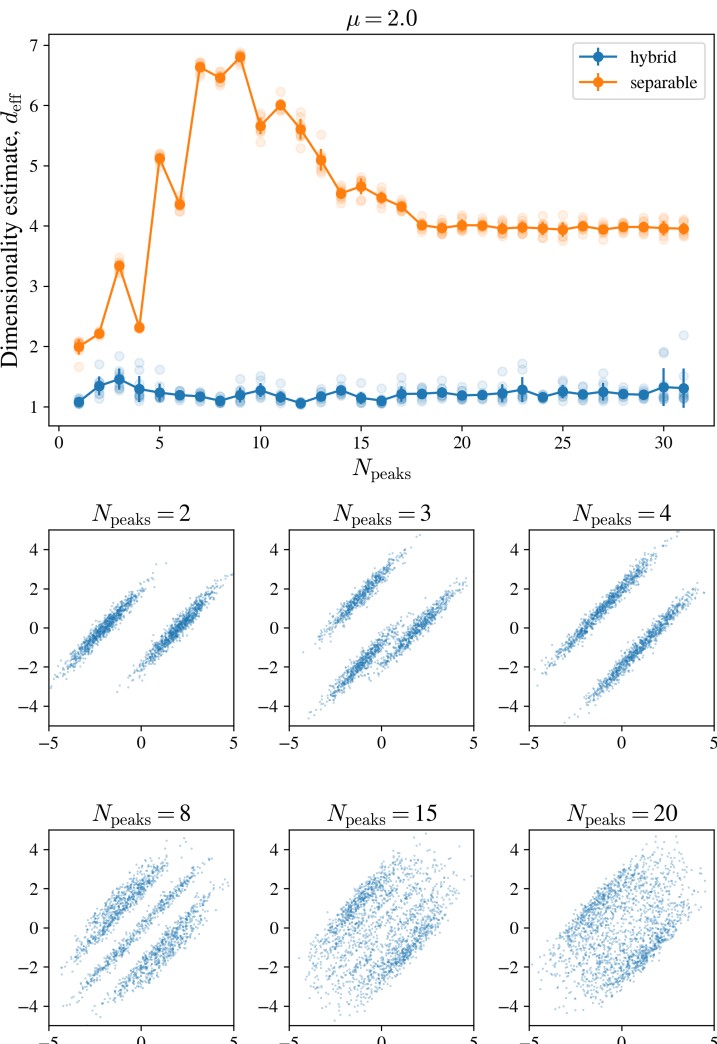

Figure 13: **Effective dimensionality of the Gaussian mixture versus number of clusters. (top)** Across mixtures with increasing $N_{\text{peaks}}$, the hybrid critic ($k_z = 64$) tracks the latent dimensionality ($K_Z = 1$), while the separable estimate shows a non-monotonic dependence on $N_{\text{peaks}}$. Under the patchwise interpretation discussed in the text, the separable dimensionality reflects the number of approximately Gaussian regions needed to represent the multimodal dependence. This is consistent with the drop in the separable estimate from $N_{\text{peaks}} = 3$ to 4 (see representative samples, **(bottom)**) and from 5 to 6 (not shown). As always, semi-transparent markers denote individual trials.

## D.2 SMALLER EMBEDDING SPACE: FEWER SAMPLES TO ESTIMATE MI

Using a hybrid critic can be beneficial even when the goal is not dimensionality estimation. For multimodal latent distributions such as our representative Gaussian mixture, a separable critic learns an effectively higher-dimensional embedding ($d_{\text{eff}} \approx 7$) than the hybrid critic ($d_{\text{eff}} \approx 1$). This increase in effective dimension directly translates into a higher sample requirement for accurate MI estimation: higher-dimensional embeddings are harder to populate, and MI estimation error grows with the effective dimensionality of the dependence structure, as argued in Abdelaleem et al. (2025a) and in related sample-efficiency analyses of simultaneous reduction methods (Abdelaleem et al., 2024) and information bottleneck objectives (Martini & Nemenman, 2024). Indeed, Fig. 14 shows that the hybrid critic reaches accurate MI estimates with substantially fewer samples than the separable critic for the

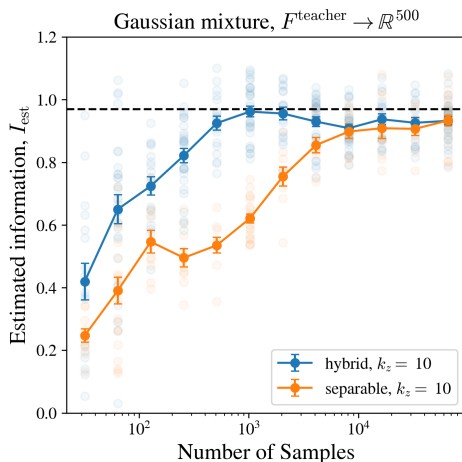

Figure 14: **Sample efficiency for MI estimation on the Gaussian mixture.** For the representative Gaussian mixture, the hybrid critic learns a lower-dimensional embedding and consequently reaches accurate MI estimates with fewer samples than a separable critic. Markers show individual trials (semi-transparent), and curves denote the mean / standard deviations across trials.

same underlying mixture distribution. We therefore conclude that the hybrid architecture can improve not only dimensionality identifiability but also practical MI estimation from limited data by reducing the effective embedding dimension.

### D.3 2D Ising model

Additional results from training on the simulated Ising-model spin configurations are shown here, including the MI as a function of temperature (Fig. 15). For $T < T_c$ (in the ferromagnetically ordered phase at $h = 0$), the MI estimate approaches $\sim 1$ bit, consistent with the two symmetry-related magnetization sectors, and it drops toward zero for $T > T_c$ in the disordered phase (Fig. 15B). We also show the unscaled dimensionality estimate $d_{\text{eff}}$ for different system sizes (Fig. 15C), displaying $d_{\text{eff}}$ only when the MI estimate is above a reliability threshold (here $I_{\text{est}} > 0.5$ bits). The peak MI, $I_{\max} = \max_T I$, and the peak location $T_{\max}$ follow the expected finite-size scaling behavior, $I_{\max} \sim \log L \sim \log N$ and $T_{\max} - T_c \sim L^{-\nu} = L^{-1}$, as shown in Fig. 15E (Goldenfeld, 2018; Bialek et al., 2001; Tchernookov & Nemenman, 2013).

### E Experimental Details: Synthetic Data

Here we provide the specific hyperparameters, architectures, and training protocols used for the synthetic benchmarks in Section 3.

### E.1 Data generation and transformations

To simulate high-dimensional observations, a low-dimensional latent variable $Z \in \mathbb{R}^{K_Z}$ is drawn from a ground-truth distribution and mapped to high-dimensional observation vectors $X, Y \in \mathbb{R}^{500}$ via frozen, randomly initialized neural networks ("Teacher" networks).

**Latent Distributions.** In the text and the appendices, we discuss the following joint or shared latent distributions:

1. Joint Gaussian: A standard baseline with total MI $I$ (in bits) and latent dimensionality $K_Z$, unit variance and zero mean. This MI is split equally across dimensions,

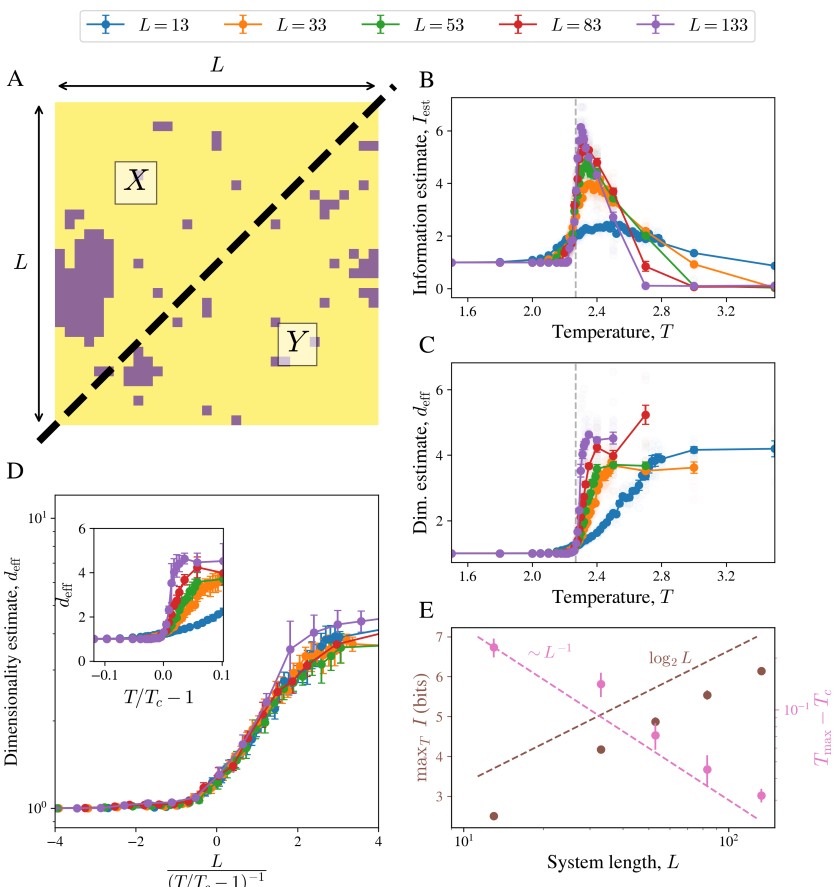

Figure 15: **Ising model: mutual information and dimensionality. (A)** Schematic illustrating a spin configuration and the spatial partitioning used to construct paired datasets. At each temperature $T$ and system size $L$, we generate 10,000 approximately independent equilibrium configurations of the 2D nearest-neighbor Ising model ($J = 1.0$, $h = 0.0$) using Markov chain Monte Carlo (see App. F.1). **(B)** The estimated MI using our MI estimator (with $k_z = 64$) peaks near the known critical temperature $T_c \approx 2.269$. **(C)** The corresponding effective dimensionality shared between the two spatial partitions. Below $T_c$ it approaches $\sim 1$, reflecting the macroscopic magnetization degree of freedom in the ordered phase, and increases as correlated domains grow near criticality. **(D)** Near criticality the correlation length diverges as $\xi \sim |T - T_c|^{-\nu}$ with $\nu = 1$, so $d_{\text{eff}} \sim L/\xi$ implies the collapse of $d_{\text{eff}}$ across $L$ when plotted against the standard scaling variable $L^{1/\nu}(T/T_c - 1)$. **(E)** From (B) we extract $I_{\max}$ and $T_{\max}$ for each $L$, with means and standard deviations obtained by bootstrapping across trials. This recovers the leading-order finite-size scalings $I_{\max} \sim \log L = \frac{1}{2} \log N$ (with $N = L^2$ and prefactor $\frac{1}{2}$ corresponding to one macroscopic degree of freedom, the magnetization (Tchernookov & Nemenman, 2013)) and $T_{\max} - T_c \sim L^{-\nu}$ with $\nu = 1$. Throughout, solid markers denote means over 10 independent training trials with error bars showing the standard deviation; in (B,C) semi-transparent markers show individual trials.

i.e. $\rho = \sqrt{1 - 2^{-2I/K_Z}}$.

$$p(z_x, z_y) = \mathcal{N}\left(\begin{bmatrix} z_x \\ z_y \end{bmatrix}; 0, \begin{pmatrix} \mathbb{I}_{K_Z} & \rho\,\mathbb{I}_{K_Z} \\ \rho\,\mathbb{I}_{K_Z} & \mathbb{I}_{K_Z} \end{pmatrix}\right). \tag{41}$$

Unless otherwise specified, we use $I = 2.0$ bits and $K_Z = 4$.

2. Gaussian Mixture: A complex, multi-modal distribution deliberately designed to be challenging with $K_Z = 1$. It has $N_{\text{peaks}}$ equally likely, unit-variance jointly Gaussian

components, with means equally spaced on a ring of radius $\mu$ in $(z_x, z_y)$ space, and per-component correlation $\rho_{\text{peak}}$:

$$p(z_x, z_y) = \frac{1}{N_{\text{peaks}}} \sum_{k=1}^{N_{\text{peaks}}} \mathcal{N}\left(\begin{bmatrix} z_x \\ z_y \end{bmatrix}; \begin{bmatrix} \mu \cos\theta_k \\ \mu \sin\theta_k \end{bmatrix}, \begin{pmatrix} 1 & \rho_{\text{peak}} \\ \rho_{\text{peak}} & 1 \end{pmatrix}\right), \quad \theta_k = \frac{2\pi k}{N_{\text{peaks}}}, \tag{42}$$

where $\rho_{\text{peak}} = \sqrt{1 - 2^{-2I_{\text{peak}}}}$. Unless otherwise specified, $N_{\text{peaks}} = 8$, $\mu = 2.0$, and $I_{\text{peak}} = 2.0$ bits, equivalent to $\rho_{\text{peak}} \approx 0.97$.

3. Noisy Hyperspherical Shell: This is used in the shared-latent setting, with $z$ sampled near a hyperspherical surface $S^{K_Z - 1} \subset \mathbb{R}^{K_Z}$ of radius $r$, with radial noise of variance $\sigma_r^2$:

$$z = (r + \epsilon_r)\,\hat{n}, \qquad \hat{n} \sim \text{Unif}(S^{K_Z - 1}), \quad \epsilon_r \sim \mathcal{N}(0, \sigma_r^2). \tag{43}$$

Unless otherwise specified, $K_Z = 3$, $r = 4$, $\sigma_r = 0.5$.

4. Swiss Roll: A standard curved manifold of intrinsic dimension 2 embedded in $\mathbb{R}^3$, generated by $t \sim \text{Unif}(t_0, t_1)$ and $h \sim \text{Unif}(h_0, h_1)$:

$$Z_1 = t \sin t, \qquad Z_2 = t \cos t, \qquad Z_3 = h. \tag{44}$$

Unless otherwise specified, $t_0 = 1.5\pi$, $t_1 = 3.5\pi$, $h_0 = 0$, $h_1 = 15$.

**The Teacher Networks.** The mapping functions $F_X, F_Y : \mathbb{R}^{K_Z} \to \mathbb{R}^{500}$ are parameterized as Multi-Layer Perceptrons (MLPs). Unless otherwise stated (e.g., the "Linear" baseline in Fig. 2B,E), these networks have one hidden layer of size 1024, with Xavier normal initialization and Softplus activation to induce nonlinear structure in the observation space. We add white noise relative to the signal strength where specified (e.g., in Fig. 3).

## E.2 CRITIC ARCHITECTURES

All critic architectures use LeakyReLU activations in all layers except the final output layer. All layers are initialized with Xavier uniform initialization.

**Separable Critic.** The separable architecture, defined as $T(x, y) = g^X(x)^\top g^Y(y)$, uses independent encoders for $X$ and $Y$. These encoders are parameterized as two-layer MLPs with 128 hidden units, mapping the input to an embedding of dimension $k_z$, which is swept (or fixed to a large value) across experiments.

**Hybrid Critic.** Our hybrid architecture, $T(x, y) = T_\theta([g^X(x), g^Y(y)])$, retains the same encoder backbone as the separable critic but fuses the embeddings via a concatenated head. This mixing head $T_\theta$ is an MLP that takes the concatenated embeddings $[g^X(x), g^Y(y)]$ as input and processes them through a single hidden layer with 64 units.

## E.3 TRAINING PROTOCOLS

All networks are implemented in `PyTorch` and optimized using Adam , a learning rate $5 \times 10^{-4}$.

**Infinite Data Regime.** This is the regime used for Figs. 2, 3, 10, and 4. Because a fresh batch is generated at every training step, there is no overfitting. Estimators are trained for 20,000 iterations with batch size 128. The reported MI is the average over the final 10% of training steps.

**Finite Data Regime.** For the finite-data experiments in Fig. 5, we train on fixed datasets with sample size $N$ as indicated in the figure. We train for 100 epochs, evaluating a held-out test set of 128 samples at each epoch. We then use the Max-Test heuristic (App. B.1), selecting the checkpoint that maximizes the MI estimate on this test set.

**Computational Resources.** Experiments were run on AWS instances of type L4, L40s, A100, and H200, using configurations appropriate to the scale of each sweep. Runtime varies with dataset size and training parameters. For example, one trial in the infinite-data resampling regime (Fig. 2) with $K = 500$ observed dimensions and 20,000 iterations at batch size 128 takes $\sim 70$ seconds, while a single-pendulum run (Fig. 6) for one trial of 200 epochs with 1,000 initial conditions (57,000 samples of size $2 \times 128 \times 128$) takes $\sim 3{,}000$ seconds.

## F    Experimental Details: Physical Systems

### F.1    2D Ising model

We generate spin configurations using Markov Chain Monte Carlo (MCMC) simulations of the two–dimensional ferromagnetic Ising model with local single–spin Metropolis updates (Metropolis et al., 1953; Newman & Barkema, 1999) on a square lattice of linear size $L$ with periodic boundary conditions. For each temperature $T$, configurations are sampled using single-spin stochastic updates that satisfy detailed balance, producing equilibrium ensembles of spin states. One Monte Carlo sweep consists of sequential updates on the two checkerboard sublattices. Multiple independent replicas are evolved in parallel by embedding several $L \times L$ systems into a larger lattice; each replica is initialized in a uniform up or down state chosen at random and evolved independently. After a fixed number of sweeps between measurements, observables are recorded separately for each replica, yielding ensembles of approximately decorrelated configurations.

Throughout this work we restrict to zero external field ($h = 0$) and unit coupling ($J = 1$). We sweep system sizes from $L = 13$ to $L = 133$ and temperatures from $T = 0$ to $T = 4.5$, spanning both ordered and disordered phases and crossing the critical region. For each $(L, T)$, we generate collections of 10,000 equilibrium configurations by running fixed blocks of Monte Carlo sweeps between measurements, and treat each replica as an independent sample. These spin configurations form the datasets used in the main text.

At each $(L, T)$, we construct paired views $(X, Y)$ by spatially splitting the lattice as shown in the main text, and train the MI estimator. The encoders and concatenated head use the same architecture as in the synthetic benchmarks (encoders: 2-layer MLP with 128 hidden units; concatenated head: 1-hidden-layer MLP with 64 units; LeakyReLU activations; Xavier initialization). Models are trained with batch size 128 for up to 100 epochs, using the max-test stopping criterion. We use 10% of samples as a test set and report $\hat{I}_{\text{train}}(t^*)$ at the selected epoch. Dimensionality is reported as the participation ratio of the singular values of the cross-covariance of the trained encoder representations at the max-test epoch.

### F.2    Pendulum dynamics

To evaluate our estimator on high-dimensional data with known physical degrees of freedom, we use the video dataset of Chen et al. (2022), focusing on the Single Pendulum and the Rigid Double Pendulum. The basic setup—constructing paired views by predicting future from past in a dynamical system—has close analogues in dynamical-systems theory (e.g., delay embeddings and data-driven dynamics) (Takens, 2006; Schmid, 2022), in predictive-information formulations (Bialek et al., 2001; Meng et al., 2022), and in modern representation-learning approaches based on predictive objectives (van den Oord et al., 2018; Mardt et al., 2018; LeCun, 2022).

**Physical Parameters.** The Single Pendulum has mass $m = 1\,\text{kg}$ and length $L = 0.5\,\text{m}$; its state is $(\theta, \dot{\theta})$, so that true $d = 2$. The Rigid Double Pendulum consists of two arms ($L_1 = 20.5\,\text{cm}$, $L_2 = 17.9\,\text{cm}$, $m_1 = 0.262\,\text{kg}$, $m_2 = 0.11\,\text{kg}$) and has state $(\theta_1, \theta_2, \dot{\theta}_1, \dot{\theta}_2)$, so that true $d = 4$.

**Data Preprocessing.** Videos are $128 \times 128$ RGB frames, denoted as $\Phi$. We convert to grayscale, so that $\Phi \in \mathbb{R}^{128 \times 128}$. To form paired views, we use time: since velocities are not determined by a single frame, we perform delayed embedding (Takens, 2006) by concatenating consecutive frames and defining $X = [\Phi_t, \Phi_{t+1}]$ and $Y = [\Phi_{t+2}, \Phi_{t+3}]$.

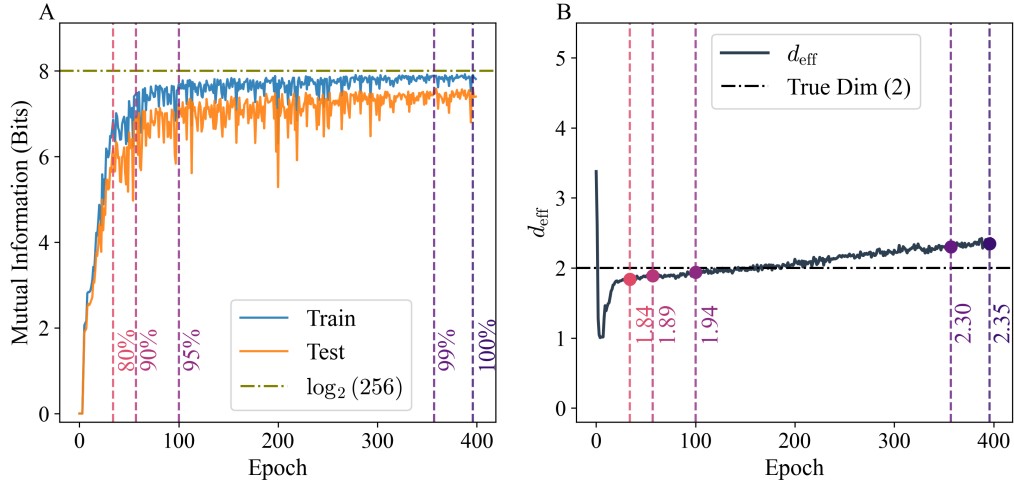

Figure 16: **Saturation and stopping in the pendulum setting.** Single-pendulum example (trained on 500 trajectories). **(A)** Train/test MI versus epoch; both approach the InfoNCE ceiling $\log_2(\text{batch size}) \approx 8$ bits. Vertical lines denote stopping points at different fractions of the maximum test MI. **(B)** $d_{\text{eff}}$ versus epoch. Although prolonged training can gradually inflate $d_{\text{eff}}$, the estimate remains near the true value ($d_{\text{eff}} \approx 2$) across a wide range of stopping thresholds.

**Dataset Organization.** Data are organized into folders, each containing 60 consecutive frames from one trajectory (one initial condition). We use 1,100 trajectories total, split into 1,000 training folders and 100 test folders. The pairing above yields 57 samples per trajectory, giving $N_{\text{train}} = 57{,}000$ pairs and $N_{\text{test}} = 5{,}700$ pairs.

**Model Architecture.** Because $X$ and $Y$ are the same system at nearby times, we enforce a Siamese constraint $g^X = g^Y$. We use the same backbone as elsewhere: an MLP encoder mapping the $2 \times 128 \times 128$ input to an embedding of dimension $k_z = 64$, and the critic head as in App. E.2, with the addition of LayerNorm in the critic to stabilize training.

**Training Protocol and Stopping.** We use Adam with batch size 256. For the full training set (1,000 folders) we train for 200 epochs and scale epochs inversely with dataset size (e.g., 400 epochs for 500 folders) to keep the number of parameter updates approximately fixed. Each epoch we evaluate MI on the full test set and on one training batch; $d_{\text{eff}}$ is computed from that training batch. Because this is a high-MI regime where InfoNCE approaches its ceiling $\log_2(\text{batch size}) \approx 8$ bits, the standard max-test rule can select $t^*$ deep in the saturation region (Fig. 16A). We therefore choose $t^*$ as the earliest epoch at which the test MI reaches a fixed fraction of its maximum. Figure 16B shows that $d_{\text{eff}}$ is stable across a broad range of such fractions, and Fig. 17 shows that the qualitative trends in $d_{\text{eff}}$ persist across thresholds and dataset sizes.

**Potential Improvements.** We emphasize that the recovery of physical degrees of freedom reported here was obtained with the same generic MLP backbone used in the synthetic benchmarks, demonstrating that the method works out-of-the-box. That said, performance in other high-dimensional modalities will likely benefit from more tailored architectures (e.g., CNN encoders to exploit spatial structure in images). In high-information regimes where InfoNCE approaches its $\log_2(\text{batch size})$ ceiling, larger batch sizes or alternative variational bounds could further reduce saturation effects. We leave such domain-specific optimizations to future work.

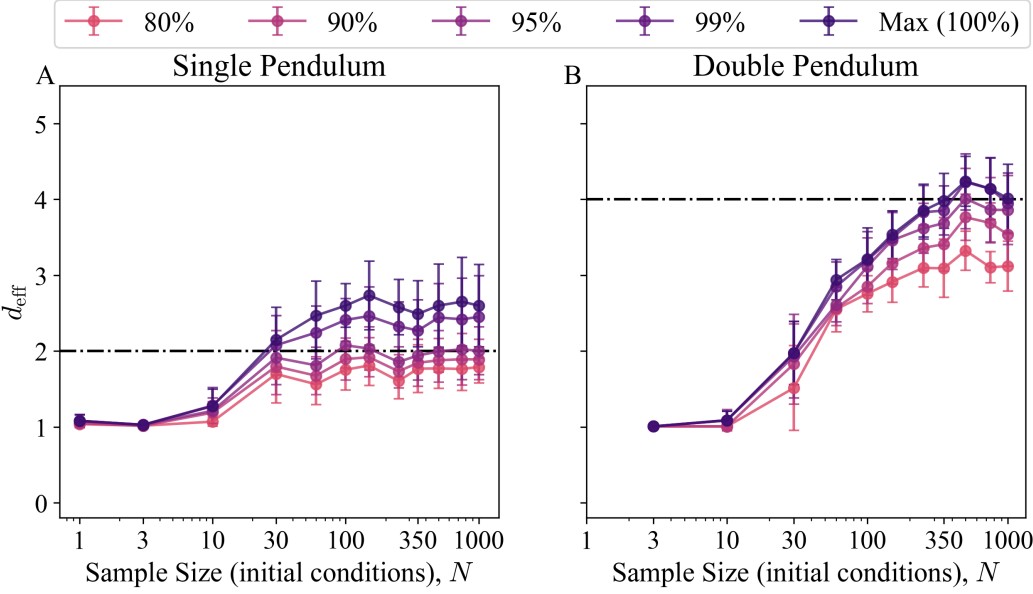

Figure 17: **Robustness to stopping thresholds.** Extension of Fig. 6B across dataset sizes (up to 1,000 trajectories) and stopping thresholds (fractions of max test MI, 80%–100%). Estimated dimensionalities are stable: $d_{\mathrm{eff}} \approx 2$ for the single pendulum (inflating toward 3 at very late thresholds) and $d_{\mathrm{eff}} \approx 4$ for the double pendulum near the maximum threshold.

