# OpenReview forum: "Mutual Information and Task-Relevant Latent Dimensionality"
_ICLR.cc/2026/Workshop/GRaM — ICLR 2026 Workshop GRaM Poster_

### Official Review · Reviewer_GsHu · 2026-02-23
**Well-motivated and technically solid contribution with convincing empirical validation**

**Rating:** 7
**Confidence:** 2

**Review:**

## Paper Summary

This paper studies the problem of estimating task-relevant latent dimensionality through a mutual-information-based framework. The authors formulate dimensionality estimation as a symmetric information-preservation problem and analyze how common separable or bilinear critics can inflate estimated dimensionality. To address this, they propose a hybrid critic architecture that decouples critic expressivity from bottleneck dimensionality and introduce a practical one-shot dimensionality estimation protocol based on participation ratios. The approach is validated on synthetic data and physics-inspired datasets, showing stable and interpretable dimensionality estimates across noisy settings.

The motivation of linking information bottleneck principles with latent dimensionality estimation is well justified, and the paper is generally well positioned within the mutual information and representation learning literature. The work clearly supports its claims through theoretical analysis and a broad set of experiments, demonstrating that the proposed hybrid critic improves robustness and mitigates dimensionality inflation observed with separable critics.


## Strengths



1. Clarity and presentation: the paper is well written, logically structured, and technically clear. The motivation, theoretical framework, and methodological design are presented in an accessible way.

2. Strong methodological insight: the analysis of dimensionality inflation caused by separable critics is insightful, and the proposed hybrid critic offers a concrete and practical solution that improves the reliability of dimensionality estimation.

4. Comprehensive empirical evaluation: the method is evaluated on synthetic experiments and real physics-inspired datasets, including noisy and controlled scenarios, which provides convincing empirical evidence supporting the claims.

5. Practical contribution: the one-shot dimensionality estimation procedure is a useful practical tool that reduces the need for repeated training runs and makes the approach more applicable in practice, aligning with GRaM's themes of scale and simplicity.


## Weaknesses

1. Dependence on MI estimation quality: as acknowledged by the authors, performance still depends on the stability of neural mutual-information estimators, which may limit robustness in more complex settings.



## Recommendation

Accept

### Key reasons:

The paper is relevant to GRaM and provides a clear, technically sound, and well-motivated contribution to understanding latent dimensionality from a geometric and information-theoretic perspective.
The hybrid critic design and one-shot estimator represent meaningful methodological improvements supported by extenditively conducted experiments.

**Pmlr Suitability:**

Yes

---

### Official Review · Reviewer_cvE8 · 2026-02-24
**Review of Mutual Information and Task-Relevant Latent Dimensionality**

**Rating:** 6
**Confidence:** 2

**Review:**

The article tackles the problem of identifying the "true" low-dimensional structure hidden in high-dimensional, noisy data, but with a twist: it focuses on dimensions that actually matter for a specific prediction task, rather than on an abstract intrinsic measure. The authors frame this as preserving mutual information (MI) through a symmetric bottleneck, which feels like a clever repurposing of existing neural MI tools. They point out flaws in standard critics (like separable ones inflating dimensions for nonlinear data) and propose a hybrid critic that mixes embeddings more flexibly without bloating the size. There's also a neat one-shot method to estimate effective dimension from an overparameterized model, and they extend it to intrinsic dims via view-splitting. Tests on synthetic and physics datasets (such as Ising models and pendulums) show it holds up better in noisy scenarios than classics like the Levina-Bickel or Two-NN methods.

**Strong points:**

On the plus side, the paper's conceptual grounding is solid—it draws on information-theoretic roots while addressing real SciML pain points, like why older estimators flop on limited data. The hybrid critic is a smart, minimal fix that improves sample efficiency (2-3x fewer samples needed), and the empirical validation is thorough: it nails known dimensions in Gaussians/mixtures and reveals cool physics insights, like critical scaling in Ising. The finite-data tricks (max-test stopping) make it practical, and it's robust to noise where others aren't. Fits the workshop's geometry theme nicely, too.


**Negative Points:**

That said, it's not groundbreaking—it builds heavily on prior MI estimators (Belghazi, Oord) and bottlenecks (Friedman, Abdelaleem), with the hybrid feeling more like a tweak than a revolution. Scope is narrow: mostly low-dim latents (up to 4), toy physics, no high-dim or real-world messy data like neural recordings. The effective dim heuristic (participation ratio) is handy but lacks a solid theory beyond Gaussians, and ablations are thin (e.g., no deep dives on architectures or alternatives like JEPA). Could overfit to benchmarks without broader tests.

**Pmlr Suitability:**

Yes

---

### Meta-Review · Area_Chair_xd2Z · 2026-02-27

**Decision:**

Accept

**Metareview:**

Well-grounded method for estimating task-relevant data manifold dimension.  However, limited novelty and small scope.

**Relevance To Proceedings:**

Yes — suitable for PMLR (long paper)

**Relevance To Workshop:**

Yes — suitable for GRaM

---

### Decision · Program_Chairs · 2026-03-02

Accept (Poster)